# Lipid regulation of hERG1 channel function

Williams E. Miranda [1], Jiqing Guo[2], Haydee Mesa-Galloso[1], Valentina Corradi [1], James P. Lees-Miller[2], D. Peter Tieleman [1✉], Henry J. Duff [2✉] & Sergei Yu. Noskov [1✉]

The lipid regulation of mammalian ion channel function has emerged as a fundamental mechanism in the control of electrical signalling and transport specificity in various cell types. In this work, we combine molecular dynamics simulations, mutagenesis, and electrophysiology to provide mechanistic insights into how lipophilic molecules (ceramide-sphingolipid probe) alter gating kinetics and $K^+$ currents of hERG1. We show that the sphingolipid probe induced a significant left shift of activation voltage, faster deactivation rates, and current blockade comparable to traditional hERG1 blockers. Microseconds-long MD simulations followed by experimental mutagenesis elucidated ceramide specific binding locations at the interface between the pore and voltage sensing domains. This region constitutes a unique crevice present in mammalian channels with a non-swapped topology. The combined experimental and simulation data provide evidence for ceramide-induced allosteric modulation of the channel by a conformational selection mechanism.

[1] Centre for Molecular Simulation and Department of Biological Sciences, 507 Campus Drive, University of Calgary, Calgary, AB, Canada. [2] Department of Cardiac Sciences, Libin Cardiovascular Institute of Alberta, 3280 Hospital Dr., University of Calgary, Calgary, AB, Canada. ✉email: tieleman@ucalgary.ca; hduff@ucalgary.ca; snoskov@ucalgary.ca

Lipophilic regulation of integral membrane proteins known as voltage-gated ion channels has been firmly established as an essential physiological mechanism[1–3]. The lipid-specific binding to ion channels from different families has shown a direct impact on the conformational equilibrium and kinetics of their gating states and ion permeation[4–6]. While lipid regulation of membrane protein function is firmly established, the mechanisms by which lipids induce functional changes in voltage-gated ion channels remain poorly understood. A major breakthrough in near-atomic resolution Cryo-EM studies led to the discovery of a novel structural topology, so called non-swapped topology, in human ion channels[7–10]. Ion channels from ERG, HCN and CNG families feature a distinctive packing of the Voltage-Sensing Domain (VSD) against the Pore Domain (PD), known as nonswapped topology. This unique topology may be linked to dramatically different permeation and gating regulatory mechanisms compared to well-studied canonical swapped domain channels[11].

Previous functional studies have linked altered gating kinetics and conduction for human ERG1 (expressed in HEK293 cells) to changes in the levels of membrane-localized molecules, such as cholesterol, polyunsaturated fatty acids, and some types of phospholipids[2]. The accumulation of sphingolipid-derived ceramides in the myocardium of patients with cardiovascular disease has been related to prolongation of the QT interval in the electrocardiogram and increased risk of developing heart arrhythmias[12,13]. The mechanisms proposed for ceramide-induced modulation of hERG1 properties include indirect effects on cell expression, altered channel trafficking to the plasma membrane, and the translocation of the channel into lipid rafts[14,15], but the molecular mechanism of hERG1 blockade by ceramides remains unknown.

In this work, we combined molecular dynamics (MD) simulations at the coarse-grained (CG) and all-atom (AA) levels of resolution using the workflow recently developed[4] with mutagenesis and electrophysiology experiments to understand the molecular mechanisms by which membrane-localized ceramides modulate hERG1 function. Specifically, we used the synthetic probe $C_6$-ceramide, hereafter referred to as ceramide or CER6 (Supplementary Fig. 1a-inset), which mimics many effects of the endogenous ceramides during physiological or pathological stimuli[14–17]. Our electrophysiology experiments show that ceramide application results in activation of the channel at more hyperpolarized potentials, faster deactivation rates, and block of $K^+$ currents ($I_{hERG1}$) similar to that produced by many common blockers associated with drug-induced QT prolongation[18]. Our MD simulations based on the recently reported hERG1 open-

state structure[10] (Fig. 1a–d) show accumulation of ceramides at the VSD–PD interface created by the nonswapped topology of the channel. In this region, the amphipathic ceramides can establish long-lasting contacts with polar, hydrophobic, and amphipathic residues. In silico mutagenesis at the VSD–PD interface shows reduced accumulation of ceramides in this region. Overall, these observations indicate that ceramides are nonconventional tonic blockers of $I_{hERG1}$ that target the channel through the lipophilic route and interact allosterically with aromatic residues in the pore domain via the crevices present in the nonswapped topology. The combination of electrophysiology and modelling provides evidence that CER6 is a gating modulator that promotes channel deactivation via a conformational selection mechanism rather than blockade of the conventional hydrophilic permeation pathway located in the pore domain.

## Results

### Effects of ceramide application on hERG1 isoform A properties.
Exogenous application of increasing concentrations of CER6 (2, 20, 100, and 500 µM) led to a progressive decrease of peak tail current for hERG1A-WT (Supplementary Fig. 1a), resulting in half-maximal inhibitory concentration ($IC_{50}^{CER6}$) and a Hill coefficient of ~22 µM and ~1, respectively (Table 1). Furthermore, apparent tonic block of the channel was observed at the highest ceramide concentration of 500 µM (Fig. 2a, b), suggesting that ceramides targeted the closed state of the channel. We also tested the inhibitory effects of ceramides with shorter and longer tails than CER6 (CER2 and CER8) using the same protocol as above. The $IC_{50}^{CER8}$ and Hill coefficient resulted ~98 µM and ~1, respectively (Supplementary Fig. 1a), showing near five-fold decrease in inhibitory potency compared to CER6. In contrast, the CER2 did not decrease WT currents to 50% even at the highest concentration of 500 µM studied (Supplementary Fig. 1a). Overall, these results indicate that CER6 is the strongest channel

**Table 1 Values of $IC_{50}$ and Hill coefficients calculated from dose-response curves for hERG1 variants.**

| hERG1 variant | WT[a] | B | Y652A[a] | F656C[a] | M651T[a] | F557L[a] |
|---|---|---|---|---|---|---|
| $IC_{50}^{CER6}$ [µM] | 22 | 8 | 17 | 385 | 46 | 33 |
| \|Hill\| | 0.8 | 1.1 | 1.0 | 0.5 | 0.9 | 1.9 |
| Stable level[b] | 0.3 | 0.1 | 0.1 | 0[c] | 0.4 | 0.4 |

[a]Reported for hERG1 isoform A.
[b]Residual current ($I_{hERG1}$/Icon) extrapolated at infinite [CER6].
[c]Estimated value. No clear asymptote was observed even at [CER6] = 500 µM.

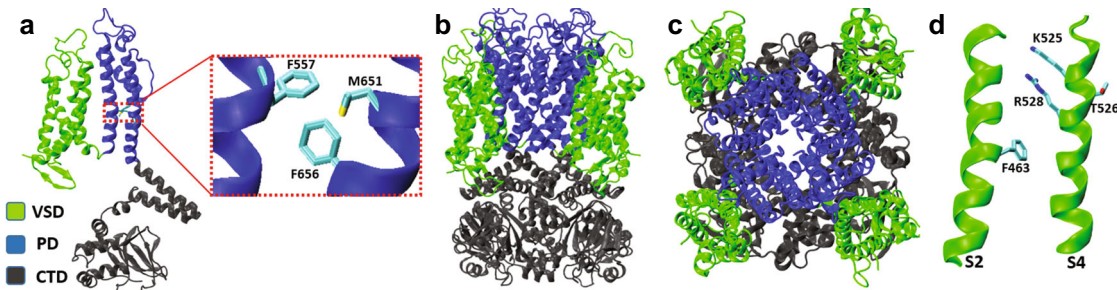

**Fig. 1 Structural domains of the open-state hERG1 channel considered in this work (PDBID: 5VA2). a** Structural organization of one monomer. The voltage-sensor domain (VSD), the pore domain (PD), and the cytoplasmic domain (CTD) are highlighted in green, blue, and gray cartoons, respectively. The inset displays key residues (licorice representation) of the PD accessible from the VSD–PD interface. The panels **b** and **c** show side- and extracellular views, respectively, of the full tetrameric hERG1 structure. Panel **d** displays the S2 and S4 helices (green cartoons) from one of the VSDs, highlighting key residues (licorice representation) involved in voltage-dependent gating. The C, O, N, and S atoms in **a** and **d** are colored in cyan, red, blue, and yellow, respectively.

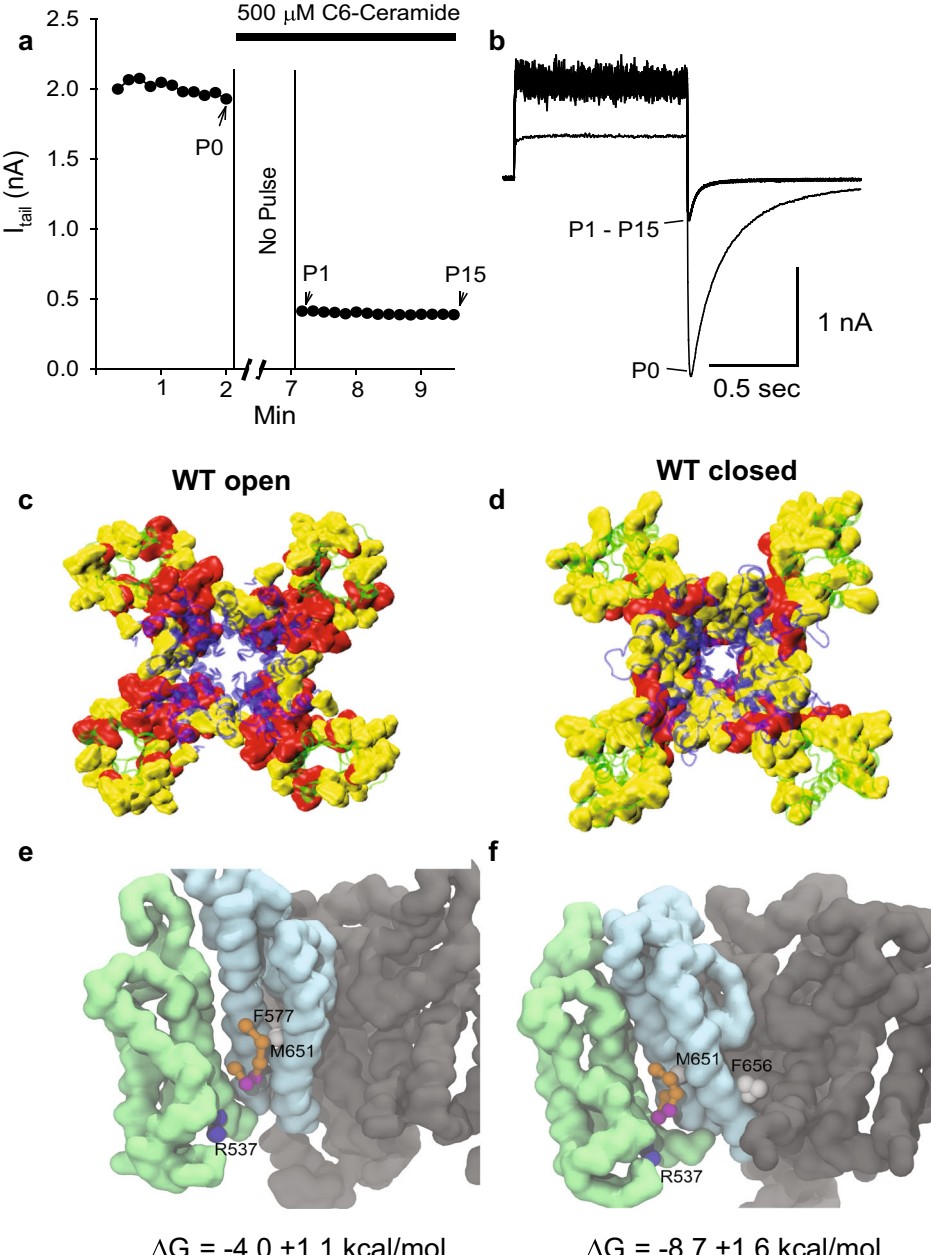

**Fig. 2 Ceramides induce apparent closed-channel block. a** Tonic block experiments on cardiomyocytes cell-line. The time course of the change of $I_{hERG1A}$ tail currents were recorded by a train of test pulses. After recorded baseline control currents (P0), 500 μM $C_6$-ceramide was added and the membrane potential was held at −80 mV with no test pulse given for 5 min. Then, the test pulses (P1-P15) were reapplied and channel currents were recorded in the presence of ceramide. **b** Superimposed current traces of P0 and P1-P15. Similar results were found in all 5 tested cells. Raw data are provided in Source Data—Fig. 2a, b. Panels **c** and **d** show the contact mapping for ceramides (extracellular view) from CG-MD simulations for hERG1A-WT open- ($n = 4$) and closed- ($n = 3$) PD states, respectively. The voltage-sensor and pore-domains (VSD and PD) are shown as green and blue cartoons, respectively. The residues identified using the maximum-occupancy criteria are shown in a red surface representation. The residues identified through contact-duration but not through maximum-occupancy metrics are highlighted as a yellow surface. A color-coded list of residues is provided in Source Data—Fig. 2c, d. For each system, the contact-duration and maximum-occupancy metrics were calculated considering the tetrameric structure of the channel, totalling $4 \times n$ replicas per-residue (see Source Data for Supplementary Figs. 5 and 6). The last 8 microseconds of each replica were used for analysis. Panels **e** and **f** show an illustrative snapshot for the binding of CER6 at the VSD–PD interface (light green and blue molecular surfaces, respectively) of the open- and closed-PD states, respectively. Only one of four VSD–PD interfaces is shown for clarity. The other subunits are shown in dark-gray color. The residues R537, F557, M651, and F656 are shown for perspective only, represented as spheres and colored according to polarity (blue: polar(+), white: apolar). The ceramide molecule is shown in balls and sticks representation and colored based on bead polarity (purple: polar headgroup, orange: apolar tails). The binding-free energy calculated with FEP approach are shown (Source Data—Supplementary Fig. 2e, f).

blocker among the tested ceramides, establishing its utility for characterizing molecular mechanisms of channel–lipid interactions.

To further characterize the effect of this lipid on conduction and kinetic properties of WT, we chose [CER6] = 20 μM, which is similar to the $IC_{50}^{CER6}$ value (Table 1). Application of ceramide concentrations in the micromolar range (1–40 μM) have been found to trigger pathophysiological stimuli for apoptosis in cardiomyocyte ischemia/reperfusion cell models[14–17] and mimics the $IC_{50}$ values of mid-affinity hERG1 blockers. For the WT, there was an ~35% reduction of peak tail current with respect to control experiments (Supplementary Figs. 1b and 2a). We also observed a left-shift of half-maximal activation voltage ($V_{1/2}^{act}$) by ~15 mV (Supplementary Fig. 2b) and a decrease of the deactivation time-constant ($\tau^{deact}$) by ~50% (Supplementary Fig. 2c). These alterations are reflected in the I–V relationship, which shows that ceramides induce activation of the channel at more hyperpolarized voltages ($-20 < V < 0$ mV), but the current is more inhibited at depolarized voltages ($V \geq +20$ mV) with respect to control experiments (Supplementary Fig. 1c). In summary, ceramide application induces a profound alteration of the $K^+$ currents and gating properties of the WT channel.

**Properties of in silico membrane models**. Next, we assessed the stability and properties of protein-free bicomponent membrane patches (POPC:CER6 = 8:2) to be used as models for investigating hERG1–lipid interactions in CG- and AA-MD simulations. Overall, we found similar values for surface area per lipid (general and specific), bilayer thickness, distribution of functional groups along the membrane normal and the absence of ceramide clusters in both CG- and AA-MD simulations (Table 2 and Supplementary Fig. 3a and b). The use of CG-MD simulations resulted in faster lateral diffusion coefficients and a slight asymmetry in the distribution of ceramide headgroups along the membrane normal. The asymmetry in ceramide distribution between the two leaflets is attributed to flip-flop events sampled during the CG-MD simulation (Table 2 and Supplementary Fig. 3a and b). Specifically, we observed a total of 18 flip-flop events from 14 (out of 100) CER6 molecules in this patch (Table 2), which showed either single (10 molecules) or double (four molecules) transitions (Supplementary Fig. 3c and d) at a rate of $3.6 \pm 1.4$ events/μs (Supplementary Fig. 3e). The increased diffusion rates in the plane of the membrane and the occurrence of flip-flop events in

CG-MD simulations are a consequence of the smoother energy surface in CG systems, which enhances the sampling compared to AA-MD simulations[19–21].

**Mapping ceramide contact regions for WT open- and closed-PD states**. We performed four replicas of CG-MD (10 μs each) for WT in the open state (Supplementary Fig. 4a and b) and used the last 8 μs for the analysis of channel-ceramide contacts. Using the contact-duration metric (sum of single-contact events), we identified a total of 69 contact residues within the upper-quartile for this channel variant, which were distributed among the transmembrane helices of the channel (Supplementary Fig. 5a). Among these 69 contact residues, 33 were also identified as maximum-occupancy (longest single-contact event) outliers, which were mainly localized at the VSD–PD interface spanning helices S4–S6 (Fig. 2c and Supplementary Fig. 6). In contrast, the remaining 36 contact residues were involved in frequent short-lived contacts with the ceramides and located away from the VSD–PD interface (Fig. 2c). Accordingly, the VSD–PD interface regions of the channel showed the highest ceramide density (Supplementary Fig. 7a), although the density pattern is not symmetric among all four interfaces. The number of ceramides in contact with the channel converged to ~$31 \pm 1$ among the four replicas (Supplementary Fig. 8a). Most of these molecules (≥90%) contacted the channel for ≤1 μs during single binding events (Supplementary Fig. 9). Notably, only one or two ceramides displayed long-lasting binding events of 4–6 μs in three of the four replicas (Supplementary Fig. 9).

At the VSD–PD interface of each monomer in WT open state, a ceramide molecule can be stabilized by interactions with polar and non-polar residues. For instance, we observed contacts between ceramides' polar headgroups and the sidechain of residue R537 from VSD-S4 helix in our CG-MD simulations (Fig. 2e). Notably, the residue T526 (Fig. 1d) was specifically targeted by ceramides' headgroups and not by POPC headgroups in CG-MD simulations, while R537 was accessible to both types of lipids (Supplementary Fig. 10). In turn, ceramides' hydrophobic tails preferentially contacted residues from the VSD-facing hydrophobic surface of the PD in the vicinity of F557, M651, and F656 (Fig. 2e). We performed backmapping[22] of specific frames from CG-MDs (see Methods) to obtain atomic-level resolution of ceramide interactions at the VSD–PD interface of WT open state. In this regard, our AA-MD simulations show that ceramides' headgroup can establish hydrogen bonds with the sidechains of residues T526 and R537 (Supplementary Fig. 11a-d). From these simulations, it is also worth noting the dynamics of residue F656. The aromatic sidechain adopts two distinguishable conformational states oriented towards the protein–lipid interface or towards the water-filled internal cavity of the channel (Supplementary Fig. 12a). This F656 sidechain dynamics is reflected by the average bi-modal distribution of the sidechains' distance with respect to the center of the internal cavity (Supplementary Fig. 12b). There is a ceramide-induced shift for F656 sidechains towards the protein–lipid interface orientation in simulations performed for WT open state (POPC:CER6 patch) in comparison to our 2-μs long AA-MD simulation of the WT channel inserted in a POPC membrane patch[23] (Supplementary Fig. 12b).

We also performed CG-MD simulations (three replicas, 10 μs each) using the WT closed-state PD model[3] to investigate the influence of the PD states on ceramide binding. For the closed-state PD, the total number and distribution of contact residues as well as the number of ceramide molecules in contact with the channel (~$29 \pm 0.3$) are reminiscent of the open state (Supplementary Figs. 5b and 8b). In contrast, the number of residues

---

**Table 2 Properties of POPC:CER6 (8:2) membrane-only systems simulated in this work (AVERAGE ± SD).**

| Property/resolution | CG simulation[a] | AA simulation[b] |
|---|---|---|
| SA/lipid[c] [Å²] | 60.0 ± 0.6 | 59.7 ± 0.7 |
| SA/POPC [Å²] | 62.5 ± 0.6 | 62.3 ± 0.8 |
| SA/CER6 [Å²] | 52.1 ± 2.2 | 51.5 ± 1.6 |
| $D_{X-Y}^{POPC}$ [× $10^{-7}$ cm²/s] | 6.6 ± 0.3 | 0.9 ± 0.2 |
| $D_{X-Y}^{CER6}$ [× $10^{-7}$ cm²/s] | 13.8 ± 0.1 | 1.0 ± 0.3 |
| Thickness[d] [Å] | 39 | 40 |
| Ceramide clusters[e] | — | — |
| Flip-flop events[f] | 18 | — |

*SA* surface area; $D_{X-Y}^{lipid}$ : Lateral diffusion coefficient calculated using lipid headgroups.
[a]The last 1.5 μs of CG-MD trajectory (5-μs long) were used for analysis.
[b]The last 250 ns of AA-MD trajectory (1-μs long) were used for analysis.
[c]General SA calculated as box area (x-y plane)/total number of lipids.
[d]Membrane thickness for CG and AA systems were calculated from the average distribution of POPC headgroups along the membrane normal (Supplementary Fig. 3a and b, respectively).
[e]Ceramide headgroups were considered for clustering analysis. Further details on the calculation of these properties is available in the "Methods" section further below. The raw data for SA and clustering calculations are provided in Source Data—Table 2.
[f]Characterized according to Ingolfsson et al.[21], using the 5 μs CG-MD trajectory (Supplementary Fig. 3c-e).

resulting in maximum-occupancy outliers at the VSD–PD interface of the closed state is reduced almost by half with respect to the open state (Fig. 2d and Supplementary Fig. 6) and there is a decrease of ceramide density at these regions (Supplementary Fig. 7b). Sampling limitations are reflected by the asymmetric density profiles (Supplementary Fig. 7a and b), which limits our assessment on the number of ceramide binders to specific open/closed pore configurations (Supplementary Fig. 8a and b). Noteworthy, free energy perturbation (FEP) simulations indicate that ceramides bind with a higher affinity to the VSD–PD interface of the closed-state PD ($-8.7 \pm 1.6$ kcal/mol, Fig. 2f) compared to the open state ($-4.0 \pm 1.1$ kcal/mol, Fig. 2e) despite the reduction of maximum-occupancy residues. This indicates that WT-ceramide complex is around 4.7 kcal/mol more stable for the closed- vs open-state PD. We found apolar and polar residues at the VSD–PD interface involved in the formation of maximum-occupancy binding pockets in the closed state of hERG1-WT (see Source Data—Fig. 2c, d), specifically L532, V533, L539, E544 (S4 helix); A561, H562, W563 (S5 helix); Y611, V612, I639, L666, Y667 (S6 helix). This is inline with the amphipathic nature of ceramide molecules.

**Ceramide-induced effects on other hERG1 variants**. To further assess the relevance of residues from the VSD–PD interface on ceramide-induced alteration of $K^+$ currents and gating kinetics of hERG1A, we performed mutagenesis experiments based on our contact analysis (see above) and previous mutagenesis data on hERG1A–drug interactions[3,24]. We also assessed the influence of ceramide on the isoform B of the channel (hERG1B). This variant is 340 residues shorter than hERG1A at the N-terminal region, showing an initial segment of 36 residues that is unique to this isoform and displays a different gating behaviour compared to hERG1A[25]. In general, the single-residue substitutions hERG1A-F557L, M651T, and F656C at the PD led to higher residual currents ($I_{CER6}/I_{control}$) with respect to WT at increasing ceramide concentrations (Fig. 3a). This is reflected by changes in $IC_{50}^{CER6}$ (Table 1), most notably for F656C (~18-fold increase). The large $IC_{50}^{CER6} = 385\,\mu M$ for hERG1A-F656C prompted us to use $[CER6] = 400\,\mu M$ (instead of 20 μM) for investigating ceramide-induced gating alterations on this variant (described further below). In contrast, the variants hERG1A-Y652A and hERG1B showed smaller residual currents (Fig. 3a and Supplementary Fig. 2a) and smaller $IC_{50}^{CER6}$ values (Table 1) with respect to WT, indicating higher sensitivity to ceramide-induced block. Most of hERG1 variants displayed similar Hill coefficient of ~1 (Table 1).

The single-residue substitutions were combined to obtain double mutants, resulting in hERG1A-variants that also show a significant increase of residual currents at $[CER6] = 20\,\mu M$ with respect to WT (Supplementary Fig. 2a). Specifically, little to no block was observed for hERG1A-F656C/F557L even at the highest ceramide concentration of 500 μM (Fig. 3a). We therefore performed CG-MD simulations (four replicas of 10-μs long each) for this ceramide-insensitive double mutant. In keeping with the lack of ceramide-induced block observed in experiments, our simulations showed a drastic reduction of maximum-occupancy residues and decreased ceramide density at VSD–PD interface for hERG1A-F656C/F557L compared to WT (Fig. 3b, c and Supplementary Fig. 7a, c). Although, the distribution of contact residues is reminiscent of the WT open state (Supplementary Fig. 5c).

Notably, the VSD residue T526 (Fig. 1d) was specifically targeted by ceramides' headgroups in our simulations (see above). Therefore, we focused on exploring the effects of different substitutions at this position on voltage-dependent activation. As expected, the equivalent substitution T526S resulted in similar

$V_{1/2}^{act}$ values with respect to WT in control and ceramide application experiments (Supplementary Fig. 2b). In contrast, the nonequivalent substitutions T526M and T526N resulted in a significant left-shift of $V_{1/2}^{act}$ with respect to WT in both assays (Supplementary Fig. 2b). Furthermore, both mutations modulate differently the effects of ceramide application with respect to control experiments, displaying a significant right-shift of $V_{1/2}^{act}$ for hERG1A-T526M but no appreciably detected effects for hERG1A-T526N (Supplementary Fig. 2b).

Overall, for most of the hERG1 variants analyzed in this work, ceramide application resulted in left-shifted $V_{1/2}^{act}$ and decreased $\tau^{deact}$ values vs control (Supplementary Fig. 2b, c), indicating simultaneous activation at more hyperpolarized voltages and faster deactivation kinetics. This is reflected by qualitative trends between $IC_{50}^{CER6}$ and changes in both gating properties relative to control: $\Delta\tau^{deact}$ and $\Delta V_{1/2}^{act}$ (Fig. 3d, e, respectively). Lastly, we performed substitutions for the VSD residue L529 as negative control in this work. This residue was not among the maximum-occupancy outliers in our CG-MD simulations for the open- nor the closed state (Source Data—Fig. 2c, d). In keeping with our simulations' predictions, the hERG1-L529A mutant showed statistically similar values of residual currents vs WT and similar $V_{1/2}^{act}$ and $\tau^{deact}$ after ceramide application with respect to control experiments (Supplementary Fig. 2a–c).

**Discussion**
Our experiments and simulations show that ceramide–channel interactions can be simultaneously modulated by PD states and accessibility to residues at the VSD–PD interface of the channel. The combined simulation and experimental evidence point to a mechanism where CER6 first partitions into the plasma membrane and then preferentially targets the VSD–PD interface in the closed-state hERG1 channels (Fig. 4). We therefore suggest that ceramide-induced channel block occurs through a mechanism of conformational selection[26] of the closed-state PD/CER6 complex, which decreases the population of open-state conducting channels (Fig. 4). The hERG1 closed-state PD structure used in this work is a model derived from EAG channel structure that may capture a preactivated state, where the pore is closed but the VSDs display a depolarized (up) configuration similar to the open state[7] (Fig. 4). Our tonic block experiments cannot distinguish between the "true" closed state with both PD and VSD in the resting (closed) conformations and presumably preactivated state captured in Cryo-EM structure. Accordingly, we cannot rule out completely the possibility of ceramide targeting either or both closed and the preactivated states of the channel. It can be concluded, however, that ceramide does exhibit preferential binding to the PD in its closed state.

In the framework of the conformational selection mechanism, the inhibitory potency of ceramides can be influenced by the changes of channel's transition rate towards the closed state (see above qualitative trends between $IC_{50}^{CER6}$ and $\Delta\tau^{deact}$). This would explain, at least in part, why hERG1A-variants with residue substitutions that induce slower (F656C) or faster (Y652A) transitions towards the closed state display low and high sensitivity to ceramide-induced block, respectively. CER6 shows high inhibitory potency for hERG1B isoform, which displays a fast transition rate towards the closed state. Rapid deactivation in hERG1B is attributed to a shorter N-terminal region with respect to hERG1A-WT[25]. The measured Hill coefficients of ~1 for hERG1A and B isoforms support a noncooperative binding mechanism, where the interactions of ceramides with one protomer have no impact on lipid recruitment for the remaining three VSD–PD interfaces.

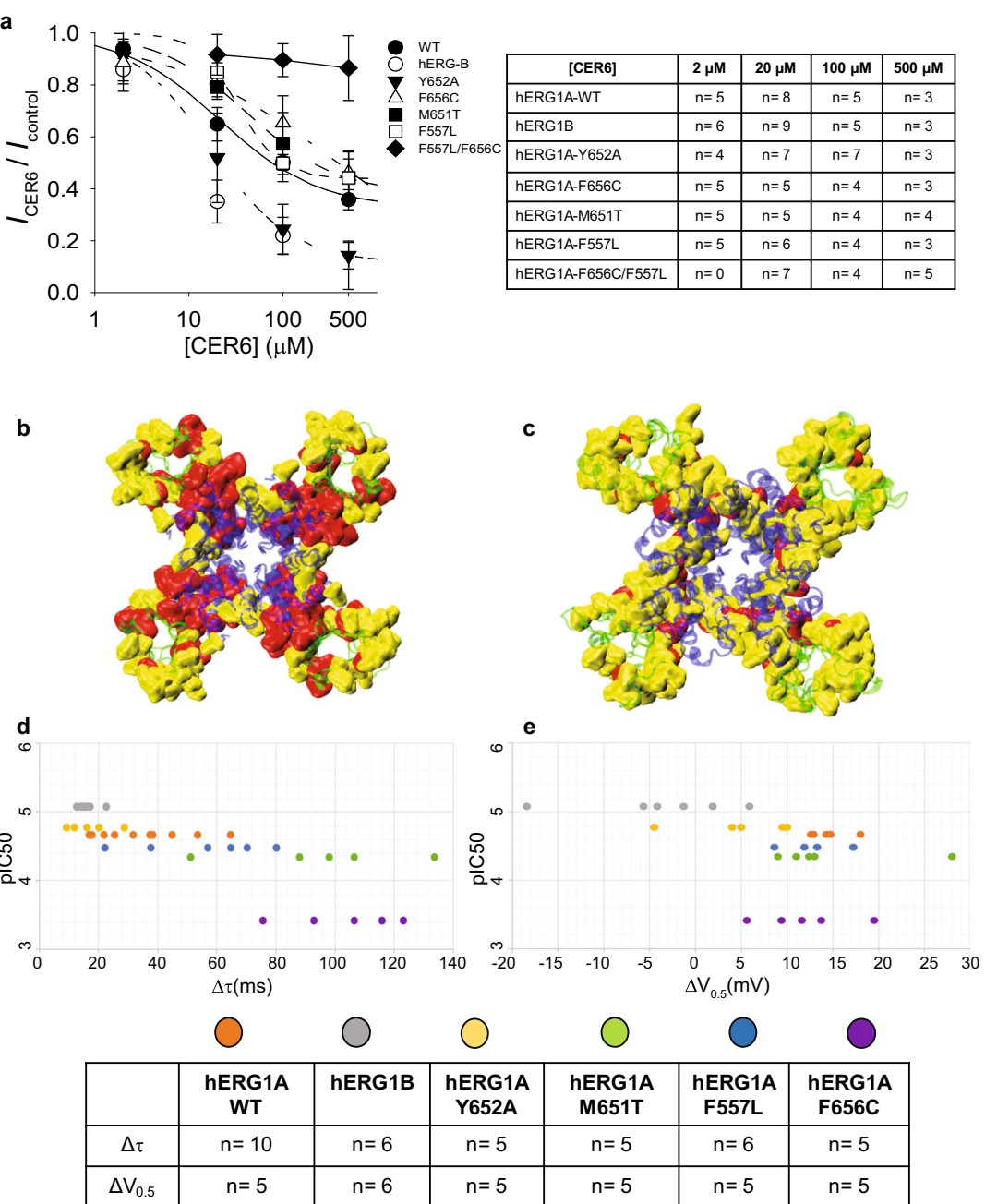

**Fig. 3 Effects of ceramide application on conduction and gating properties of hERG1 variants expressed in mammalian HEK cells. a** Dose-response relationship for increasing [CER6] = 2, 20, 100, and 500 µM. Data are represented as mean ± SD. Raw values for residual current are provided in Source Data—Fig. 3a. Panels **b** and **c** show the contact mapping for ceramides calculated from CG-MD simulations for hERG1A-WT and hERG1A-F656C/F557L open-state (n = 4 for both), respectively. The extracellular view of the channel is shown, where the VSDs and the PD are shown as green and blue cartoons, respectively. The residues identified using the maximum-occupancy criteria are represented with a red surface. The residues identified through contact-duration but not through maximum-occupancy metrics are highlighted as a yellow surface. A color-coded list of residues is provided in Source Data—Fig. 3b, c. For each system, the contact-duration and maximum-occupancy metrics were calculated considering the tetrameric structure of the channel, totalling 4 × n replicas per-residue (see Source Data—Supplementary Figs. 5 and 6). The last 8 microseconds of each replica were used for analysis. Panels **d** and **e** show scatter plots for pIC50 (−log[IC$_{50}$]) vs the change (Δ = Control − Cer) in deactivation τ and activation V$_{0.5}$ after ceramide application, respectively (see Source Data—Supplementary Fig. 2b, c). The scatter plots for pIC50 vs τ and activation V$_{0.5}$ after ceramide application are shown in Supplementary Fig. 13a, b, respectively. Deactivation τ values in **d** were measured at −100 mV, except for hERG1B (−50 mV). [CER6] = 20 µM was used for all measurements shown in panels **d** and **e**, except for the mutant hERG1A-F656C (400 µM). The legend for scatter plots for **d** and **e** is shown at the bottom: hERG1A- WT (orange), Y652A (yellow), M651T (green), F557L (blue), F656C (purple), and hERG1B (gray). The tables in **a** and **d, e** indicate the number of independent experiments (n), each performed on a different cell.

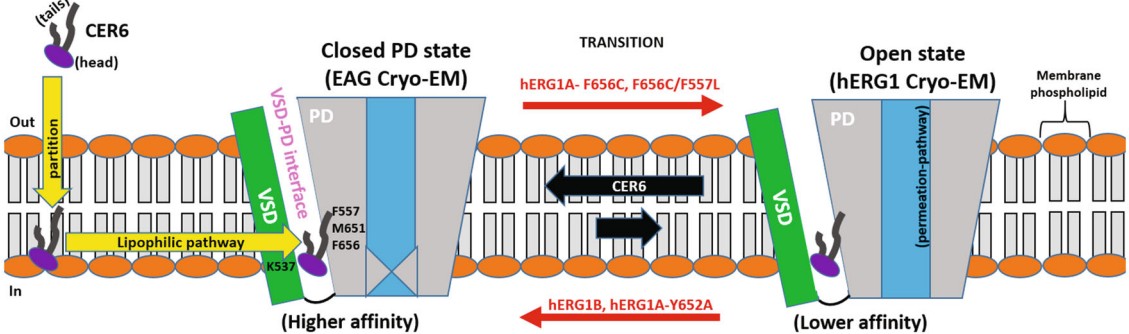

**Fig. 4 Ceramides induce hERG1 block by a conformational selection mechanism through a lipophilic pathway.** CER6 molecules partition into the membrane (after their application to the external milieu), followed by preferential binding to the interface between the voltage-sensing domain (VSD, green color) and the pore domain (PD, gray color) of hERG1 channels displaying a closed-state PD (only one of the four VSD–PD interfaces is shown). As a result, ceramides shift the population of hERG1A-WT channels towards the closed state (conformational selection), accelerating channel deactivation kinetics (left-pointing black arrow). The mutations hERG1A-F656C and F656C/F557L (right-pointing red arrow) affect ceramide interactions at the VSD–PD interface and induce slower channel deactivation kinetics, favoring the open state of the channel and the occurrence of residual currents (lack of complete channel block). In contrast, the mutation hERG1A-Y652A and the truncated N-terminal region in hERG1B isoform (left-pointing red arrow) induce faster deactivation kinetics, resulting in a higher population of ceramide-bound channels with closed-state PD and the lack of residual currents. Only two states are represented in this mechanism, corresponding to the two hERG1 channel structures used for MD simulations in this work: closed-state PD (homology model based on EAG Cryo-EM structure[7], left) and open-state PD (hERG1 Cryo-EM structure[10], right). Transitions at the VSD during activation and deactivation are not represented due to current structural-model limitations (see discussion). The ion permeation pathway in the PD of the channel is shown in blue color. Generic plasma-membrane phospholipids' headgroups and tails are colored in orange and gray colors, respectively.

The conformational selection mechanism could also help explain residual currents present in dose-response curves of hERG1A-WT, F557L, M651T, and F656C. The slow deactivation kinetics in these variants may result in a considerable population of conducting open-state channels present, even at the highest ceramide concentration. In contrast, the near-absence of residual currents in fast-deactivating hERG1A-Y652A may result from a reduced population of conducting channels at increasing ceramide concentrations. This mutation has, in general, suppressed hERG1A block by a vast array of Class III antiarrhythmic and other torsadogenic proarrhythmic drugs. Many of these proarrhythmic compounds are proposed to bind into the internal cavity of the open-state PD through the hydrophilic route[27].

Our data clearly show that ceramide-induced current inhibition of hERG1A was significantly impaired in experiments by F656C and F656C/F557L mutations. The F656C/F557L double mutant significantly affected the binding of ceramides to the VSD–PD interface in our simulations and is virtually insensitive to CER6 application in the range of studied concentrations. Although residues F656 from the S6 helices have been reported as critical for drug access to the internal cavity through the hydrophilic route[28], our AA-MD simulations show that F656 side-chains alternate configurations between the water-filled internal cavity and the channel–lipid interface. The modelling data shows that F656 could be exposed to interactions with lipid-localized probes, in agreement with the interaction mechanism recently proposed for the antiarrhythmic drug ivabradine[3]. Moreover, our simulations and experiments suggest that the modulation of ceramide-induced block by F656 is strongly related to changes in deactivation kinetics, where the substitution F656C leads to slower deactivation and a reduced number of closed-state channels that ceramides can interact with. The modelling and experimental results taken together suggest that F656, besides modulating channel deactivation kinetics as observed in our mutagenesis experiments, could potentially be involved in direct interactions with ceramides. Apparently, the residue F656 is pivotal in mediating hERG1A-ceramide interactions either directly via interactions at the VSD–PD interface or indirectly by modulation of gating transitions. All in all, these results further support our proposed mechanism where ceramides mainly interact with the membrane-facing surface of the channel via the lipophilic route and not at the internal cavity of the channel (Fig. 4).

In general, ceramides induce the speeding of deactivation and shifts in $V_{1/2}^{act}$ simultaneously. Therefore, the cause-effect relationships cannot be clearly evaluated from correlation analysis for each property alone. A decade ago, the work of Ganapathi et al.[15] suggested that ceramide-induced left-shift of $V_{1/2}^{act}$ in hERG1 may be a consequence of channel sequestration into lipid rafts[15]. Our experiments and simulations suggest that the specific targeting of T526 by ceramides could drive, at least in part, the activation of hERG1 variants at more hyperpolarized membrane potentials. This residue is flanked by K525 and R528 (Fig. 1d), two major gating-current contributors in hERG1[29]. Only the S526 substitution (among T526X mutants) shows ceramide-induced left-shifting of $V_{1/2}^{act}$ statistically similar to WT. This indicates that the hydroxyl group at this position plays a major role for ceramide (polar)interactions at the VSD as suggested by our MD simulations. Overall, our experiments show that the polarity and size of the sidechain at position 526 directly influences the effect of ceramides on $V_{1/2}^{act}$ in the channel. We also propose that R537 from S4 helix, although not a major contributor to gating current[29], could help stabilize ceramide binding at the VSD–PD interface of the channel.

In summary, our results show that ceramides are nonconventional inhibitors of hERG1. Lipid-localized CER6 exhibits preferential binding to the closed state of the channel via a conformational selection mechanism and acts as a gating modulator rather than direct blocker of the hydrophilic permeation pathway. At the molecular level, the amphipathic nature of CER6 drives its accumulation at the VSD–PD interface created by the nonswapped topology of hERG1 channel. These findings shed light on the mechanisms by which lipophilic molecules such as lipid second-messengers and hydrophobic drugs may contribute to the onset of cardiac arrhythmias.

## Methods

**Molecular biology**. Methods for site-directed mutagenesis have been previously reported[24,30]. The *hERG1* constructs were transfected into mammalian HEK cells. Single- and double-mutant constructs of *hERG1* were produced using conventional overlap PCR with primers synthesized by Sigma Genosys (Oakville, Ontario, Canada) and sequenced by Eurofins MWG Operon (Huntsville, AL) (Supplementary Table 1). Constructs were linearized with XbaI restriction endonuclease and cRNA was transcribed in vitro using the mMessage mMachine T7 Ultra cRNA transcription kit (Ambion, Austin, TX).

**General setup for electrophysiological recordings**. The extracellular solution contained (in mM) NaCl 140, KCl 5.4, CaCl₂ 1, MgCl₂ 1, HEPES 5, glucose 5.5. The pH of the solution was adjusted and kept at 7.4 with NaOH. Micropipettes were pulled from borosilicate glass capillary tubes on a programmable horizontal puller (Sutter Instruments, Novato, CA). The pipette solution contained the following: 10 mM KCl, 110 mM K-aspartate, 5 mM MgCl₂, 5 mM Na₂ATP, 10 mM EGTA-ethylene glycol-bis(-aminoethyl ether)-N,N,N,N tetra-acetic acid, 5 mM HEPES, and 1 mM CaCl₂. The solution was adjusted to pH 7.2 with KOH. Standard patch-clamp methods were used to measure the whole-cell currents of hERG1 mutants expressed in HEK293 cells using the AXOPATCH 200B amplifier (Axon Instruments)[31]. Unless otherwise indicated, the tail currents were recorded when the voltage was returned to −100 mV from +50 mV. Transfected HEK cells were patched to record the hERG1 currents[31]. For experiments evaluating tonic vs. use-dependent block, we used stable-expressing hERG1-WT cells derived from cardiomyocytes (a kind gift from Drs. Rakesh Bhat and Michael Houghton)[32]. This cell-line has the advantage of displaying very little endogenous repolarizing membrane currents.

**Voltage-dependence of activation and deactivation**. From a holding potential of −80 mV cells were depolarized for 1 s to a range of voltages from −100 to +40 mV followed by a step to −100 mV (1 s) to record the tail currents. The isochronal tail current–voltage plots were fit to a single Boltzmann function (1):

$$I/I_{max} = 1 / \left( 1 + \exp\left[ \left( V_{1/2} - V_m \right)/k \right] \right) \quad (1)$$

Where $I/I_{max}$ is the normalized current, $V_{1/2}$ is voltage of the half-maximal activation, $k$ is the slope factor and $V_m$ is the membrane potential. Deactivation of hERG1 tail currents was measured by activating channels at +40, followed with a short (5 ms) repolarization step to −120 mV and deactivating step at −120, −100, −60, and −40 mV. Currents at different voltages were normalized and fitted. The fitted data were averaged, and the n values are presented in Figures' captions. Ceramide ($C_2$, $C_6$ and $C_8$) solutions were prepared according to Bai et al.[16]. Briefly, ceramides were dissolved in DMSO in 20 mM stock solutions. Before the experiments, the ceramides stock solutions were put and vortexed in the extracellular solution containing BSA (bovine serum albumin, BSA: DMSO = 1:1). The control solutions had equivalent concentrations of DMSO and BSA.

**Statistical analysis of electrophysiological experiments**. The software Clampfit v8.2 and v9.2 and Excel 2016 were used for e-physiology data analysis. Electrophysiology data were displayed using the software SigmaPlot v14.0 and Clampex v9.2 was used to process data signal in e-physiology experiments. The null hypothesis of this study predicted no difference between the IC₅₀, $V_{1/2}^{act}$ and $\tau^{deact}$ values of the single to the double mutations assessed. The null hypothesis was rejected when the p-value was <0.05 as evaluated by a one-way Analysis of Variance with Tukey test to compare hERG1 variants. Two-tailed t-tests were performed to assess statistical differences in $V_{1/2}^{act}$ and $\tau^{deact}$ properties (ceramide vs control) for each hERG1 variant considered in this study. All variance measures for electrophysiological data are shown as mean ± standard deviation (SD). The study was exploratory, there was no a priori reason to consider whether there were additive or subtractive interactions. In addition, we acknowledge that there may be other mutations, unknown to us at this time, which could be relevant. All of the comparisons were prespecified, and all of the comparisons are reported. The n values for each point of the concentration-response relationship are presented in Fig. 3a.

**Simulation systems setup**. In this work, we considered the transmembrane and C-terminal regions (residues 398–862, Fig. 1a–d) of the open-state Cryo-EM structure of hERG1 (PDBID: 5VA2, 3.8 Å resolution)[10]. For the closed state we used the recently reported structural model (residues 405–670)[3], based on the Cryo-EM structure of the homologous rat EAG1 channel (PDBID: 5K7L, 3.8 Å resolution)[7]. The PDB Reader module of CHARMM-GUI[33] was used for the generation of the tetrameric structures and to perform in silico mutagenesis for the open state.

**Coarse-grained (CG) molecular dynamics (MD) simulations**. To provide initial mapping of ceramide distribution around hERG1 variants we performed multiple coarse-grained MD simulations using the Martini force field, version 2.0 for lipids, 2.2 for proteins and with the nonpolarizable water model[34]. The Martini–Maker module[35] implemented in CHARMM-GUI was used to build the systems in

coarse-grained (CG) representation. The proteins were embedded in a 20 × 200 Å bicomponent membrane patch composed of POPC and DPCE (8:2), resulting in 920 and 230 molecules, respectively (Supplementary Fig. 4a and b). The DPCE is a general model ceramide lipid corresponding to atomistic C(d16:1/16:0) N-palmitoyl-D-erythro-sphingosine or C(d18:1/18:0) N-stearoyl-D-erythro-sphingosine. The POPC lipid corresponds to atomistic C(16:0/d18:1) 1-palmitoyl-2-oleoyl-glycero-3-phosphocholine. The protein/membrane systems were solvated and counter-ions were added to achieve electro-neutrality at the physiological concentration (150 mM). Afterwards, the structure and topology of DPCE lipids were modified through deletion of beads C2B–C4B, which corresponds to 10–12 carbon atoms, to obtain and parameterize CER6 molecules [C(d18:1/6:0) N-hexanoyl-D-erythro-sphingosine]. The standard parameters defining bonded and nonbonded interactions for DPCE in Martini force field were then applied to the corresponding beads in the CER6. For proteins, the ElneDyn elastic network model[36] was applied to each monomer of a given tetramer. The membrane-only system was built using a similar protocol, resulting in a bilayer patch of 125 × 125 Å comprised of POPC: CER6 (8:2), with 400 and 100 molecules, respectively.

The CG-MD simulations were run using GROMACS (v5.0.7)[37]. We used restraints for the z position of the phosphate groups in the bilayers and for the positions of backbone and sidechain atoms in the protein during staged equilibration steps, according to the default protocol implemented in CHARMM-GUI. The temperature was set to T = 310 K using the velocity rescaling method[38]. The pressure was set semi-isotropically to 1 bar using the Berendsen barostat[39] with a time coupling constant of 5 ps and a compressibility value of $3 \times 10^{-4}$ bar⁻¹ (for minimization and equilibration phases) and the Parrinello–Rahman barostat[40] with a time coupling constant of 12 ps and a compressibility value of $3 \times 10^{-4}$ bar⁻¹ (for production phase). Nonbonded interactions were used in their shifted form with electrostatic interactions shifted to zero in the range of 0–11 Å and Lennard–Jones interactions shifted to zero in the range of 9–11 Å. A timestep of 10–20 fs was used with neighbour lists updated every 10 steps. Periodic boundary conditions were used in all simulations. Each production phase replica for hERG1-membrane systems was run for 10 μs (5 μs for the membrane-only system).

**Binding-free energies**. The Free Energy Perturbation (FEP) method was used. The channel-ceramide-bound systems obtained from the CG-MD simulations (Fig. 2e and f) were used as the input for the FEP calculations. Coulombic and Lennard–Jones interactions of the target lipid CER6 bound to the VSD–PD interface were fully decoupled along a coordinate in the chemical space, termed $\lambda$. To keep the CER6 molecules in the binding site and within the correct plane and orientation of the membrane at high values of $\lambda$, we followed a CER6-restraining scheme described by Corey et al.[41]. $\Delta\Delta G_{bind}$ was calculated from the energy required to decouple the CER6 molecule in the bound and free state, in each case perturbing the Lennard–Jones interactions over 19 windows (with 0.05 spacing from 0 to 0.7, and 1.0 spacing from 0.7 to 1.0). Three independent production simulations of 250 ns were run with randomized initial velocities, using a stochastic dynamics integrator. Energy values were calculated on the final 200 ns of simulation data. Simulations were run in GROMACS (v5.0.7)[37].

**All-atom (AA) molecular dynamics (MD) simulations**. The membrane-only system was built using POPC and CER160 lipids (8:2), resulting in a bilayer patch of 125 × 125 Å containing 400 and 100 molecules of each lipid type, respectively. The CER160 is a model ceramide lipid defined in the Membrane–Builder module of CHARMM-GUI[42], which corresponds to C(d18:1/16:0) N-palmitoyl-D-erythro-sphingosine. The system was solvated and counter-ions were added to achieve electro-neutrality at the physiological concentration (150 mM). Afterwards, the structure and topology of CER160 lipids were modified through deletion of C7F–C16F carbon atoms to obtain and parameterize atomistic CER6 molecules. The standard parameters defining bonded and nonbonded interactions for CER160 in CHARMM C36 force field[43] were then applied to the corresponding atom types in the CER6.

To build protein-membrane systems, we first chose five frames from one CG-MD replica of WT open state, where at least one ceramide molecule was in contact with key residues such as T526 and R537 from the VSD and residues F557, M651, and F656 from the PD. These frames were the starting points for the backmapping process using the backward.py[22] protocol. The interactions were defined by the CHARMM C36 force field for lipids and proteins[43,44]. The backmapped protein/membrane systems were resolvated in 150 mM of KCL aqueous solution with TIP3P water model[45]. The final systems' sizes were 193 × 193 × 148 Å. The atomistic simulations of membrane/protein systems (minimization, equilibration, and production phases) were run using GROMACS (v5.0.7)[37]. We used restraints for the z position of the phosphate groups in the bilayers and for the positions of backbone and sidechain atoms in the protein during staged equilibration steps, according to the default protocol implemented in CHARMM-GUI. The pressure was set semi-isotropically to 1 bar using the Berendsen barostat[39] with a time coupling constant of 5 ps and a compressibility value of $3 \times 10^{-4}$ bar⁻¹ (for minimization and equilibration phases) and the Parrinello–Rahman barostat[40] with a time coupling constant of 12 ps and a compressibility value of $3 \times 10^{-4}$ bar⁻¹ (for production phase). Electrostatic interactions were computed using the particle mesh Ewald (PME) method[46]. The LINCS method[47] was applied to restrain all the bonds, allowing a 2 fs integration timestep. The temperature was set

to T = 310 K using the velocity rescaling method[38], where the protein, the membrane, and the solute were coupled separately. The pressure was set semi-isotropically to 1 bar using the Parrinello–Rahman barostat[40] with a time coupling constant of 12 ps and a compressibility value of $3 \times 10^{-4}$ bar$^{-1}$. Periodic boundary conditions were used for all simulations. Each production phase replica for hERG1-membrane systems was run for 500 ns (1 µs for the membrane-only system).

**Contact analysis**. From the CG simulations, we assessed ceramide–protein and phosphatidylcholine–protein interactions on a per-residue basis using contact-duration and maximum-occupancy metrics, as described by Barbera et al.[48]. The former quantifies the total amount of time individual amino acid residues are within 6 Å of any lipid molecule bead. The latter is a measure of the longest continuous contact a given amino acid residue has with any lipid molecule. The contact cutoff of 6 Å is commonly used for determining contacts between beads in CG-MD simulations[48]. As hERG1 is a homotetramer, each replica of CG-MD provides by default four contact metric values for each residue. For each open-state variant (WT and mutants) and the closed-state WT system we performed four and three CG-MD replicas, respectively. Hence, the per-residue contact metric values we report here are averaged from among 16 and 12 per-residue values for open- and closed-state systems, respectively.

We also analyzed statistical significance based on percentiles as in Barbera et al.[48]. In short, we focused our analysis on residues that belonged to the upper-quartile (within the upper 25 %) of the contact-duration metric (see example in Supplementary Fig. 5a–c). We also focused on those residues that resulted in outliers from the maximum-occupancy metric (>Q3 + 1.5*IQR), where IQR = Q3 − Q1. The Q3 and Q1 are the third and first quartiles, respectively; IQR is the interquartile range; and Q3 + 1.5*IQR is the upper boundary in the boxplot (see example in Supplementary Fig. 6).

**Density calculations**. The average density of lipid headgroups, linker regions, and hydrocarbon tails along the membrane normal (Z-axis) for CG and AA membrane-only systems was calculated using the GROMACS tool g_density. We also used the VMD plugin VolMap[49] to compute 2D average density maps from CG-MD simulations using the last 8 µs of each trajectory.

**Surface area (SA) calculations**. We calculated general (SA/lipid) and specific (SA/POPC and SA/CER6) values for CG and AA membrane-only systems (Table 1). The former was obtained as the quotient of the membrane surface area in the x–y plane divided by the total number of lipids (POPC and CER6) per-leaflet. The latter was calculated using the software Grid-MAT(v2.0)[50], considering specific beads from each lipid type (PO4 and GL1 for POPC; AM1 and AM2 for CER6) and using 200 × 200 grid points with 1 Å spacing to generate tessellation-like data.

**Calculation of diffusion coefficients**. The lateral diffusion coefficient for each lipid type ($D_{X–Y}$) was calculated from the mean square displacement (MSD) of the molecules in the membrane plane after correction of periodic boundary effects:

$$\text{MSD} = \left| r(t - t_0) - r(t_0) \right|^2 \quad (2)$$

where $\left| r(t - t_0) - r(t_0) \right|$ is the distance travelled in the lateral directions by the headgroup from each lipid type between the times $t$ and $t_0$ The symbol $\langle \rangle$ denotes the average over both time and the number of molecules. The interval between the 10 and 90% of the trajectory was used to fit the MSD curve (y = 4Dt + c) to calculate the $D_{X–Y}$ using the GROMACS tool g_msd.

**Clustering analysis**. We analyzed clustering events for ceramides using the radial distribution function (rdf) built-in script from VMD(v1.9.3) and considering corrections for periodic boundaries. Specifically, we considered the formation of a ceramide cluster if two or more ceramide headgroups were found within a radius of 5 Å in the x–y plane as described by Wang and Klauda[51].

**Hydrogen bonds analysis**. Hydrogen bonds were computed using the GROMACS tool g_hbond that considers distance and angle cutoffs of 3.5 Å and 30°, respectively, for donor–acceptor pairs.

**Visualization**. The molecular graphics viewer VMD(v1.9.3)[49] was used for visualization.

**Reporting summary**. Further information on research design is available in the Nature Research Reporting Summary linked to this article.

## Data availability

Data supporting the findings of this manuscript are available from the corresponding authors upon reasonable request. A reporting summary for this Article is available as a Supplementary Information file. Source data are provided with this paper.

## Code availability

The scripts for analysis of protein–lipid interactions, clustering, surface occupancies as well as input files for Free Energy Perturbations are available from https://doi.org/10.6084/m9.figshare.13661033.v1

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

## Acknowledgements

This work was supported by the Canadian Institutes of Health Research Project Grant FRN-CIHR: 156236 (S.Y.N., H.J.D., D.P.T., and J.P.L.-M.); the National Institutes of Health grant R01HL128537-01 (S.Y.N., J.Q., and W.E.M.) and Discovery grants from the Natural Sciences and Engineering Research Council of Canada (to H.J.D and D.P.T.). W.E.M. would like to acknowledge support from a Vanier Canada Graduate scholarship, a Killam scholarship and an Alberta Innovates Health Solutions studentship. D.P.T. acknowledges further support from the Canada Research Chairs program. Calculations were carried out on Compute Canada resources, funded by the Canada Foundation for Innovation and partners. We would like to acknowledge Drs. Michael Houghton and Rakesh Bhat (Li Ka Shing Applied Virology Institute, University of Alberta) for kindly providing the hERG1-WT expressing cells derived from cardiomyocytes.

## Author contributions

W.E.M., V.C., D.P.T., and S.Y.N. designed the simulation studies. W.E.M. conducted the molecular dynamics simulations and developed methods for analysis with assistance from V.C. H.M.G. provided force field parameters for CG model of CER6 and performed FEP calculations. J.Q., J.P.L.-M., and H.J.D. designed, conducted, and analyzed the experiments. W.E.M. drafted the paper with assistance from all co-authors.

## Competing interests

The authors declare no competing interests.
