## [Peer Review File · Nature Communications]

Reviewer #1 (Remarks to the Author):

Here the authors report insights into the molecular mechanism of ceramide inhibition of hERG channels. The authors find that ceramide shifts the activation curve to more hyperpolarized voltages, but decreases the current at depolarized voltages and decreases the open time. In simulations, they find that ceramide is concentrated at the interface between the voltage sensor domain and the pore domain and makes contact with residues at this interface. Mutations of some of these residues reduces the effect of ceramide, supporting a role for this interface for ceramide inhibition of hERG channels. I have only some comments about the presentation and details about the mechanism. The importance of hERG channels to human diseases makes this an important contribution to the field.

Major Comments.

1. Pg. 4. The authors state that the mutations of T527 have no effect on ceramide-induced block, but what about effects on the ceramide-induced V shift? Or ceramide affinity? If you later want to claim that T527 is important for ceramide V shift, then you must show some data supporting that claim. Table 3 only shows $V_{0.5}$ due to mutation, not changes to ceramide-induced $V_{0.5}$ shifts.
2. Pg. 5. Not sure I understand the proposed mechanism? Why would mutations that have lower affinity for ceramide stabilize the open state (τ increased) or vice versa?
3. Pg. 5. And why would fast deactivating channels, such as hERG1B, have higher affinity for ceramide, if these contact residues are less exposed? Shouldn't it be the opposite? If a residue is less exposed then ceramide would have less opportunity to bind to this residue and the affinity should therefore be lower.
4. Fig. 5. I don't get this model. B seems to violate micro-reversibility, but I think the two arrows mean different things? The left arrow probably means that the GV is shifted left, but the right arrow means that closing is faster. So the arrows have different meanings. A better model/picture is needed to better explain this in a self-consistent manner.

Minor comments

1. Introduction. "results in activation of the channel at more depolarized potentials" should be at more hyperpolarized potentials.
2. Fig. 2. Were the mutations introduced in Herg1A or Herg1B background? Also, please mention the difference in amino acid sequence or major structural difference between 1A and 1B.
3. Pg. 3. Please describe here what you mean by "contact duration" and "maximum occupancy" in simple words. It is not that difficult and it would make it a lot easier to read the rest of the text if this is explained here.
4. Fig. 3. "Notably, T526 was specifically targeted by ceramides and not by POPC headgroups in CG-MD simulations due to the ability of ceramides to flip-flop, while R537 was accessible to both types of lipids (Supplementary Figure 8)." Only part of this is shown in Supplementary Fig 8. Fig 8 shows only that T526 is not contacted by POPC...
5. Pg. 5. Do you mean "mutations of high occupancy residues"? Use this instead of "variants with substituted high occupancy residues".
6. Suppl. Fig 7. What does maximum single-contact event mean? You mean longest single-contact event? I would rename this for clarity.
7. Suppl Fig 7. "per ceramide molecule" could be misunderstood, which in my mind means an average (e.g. per capita income). I would call it "A histogram of the longest single-contact event for

each ceramide molecules".

8. Suppl. Fig. 9A. Why is the amplitude 4? Due to 4 subunits? Or 4 ceramides interacting with one T526?

9. Suppl Fig. 10. Why red and green dots in panel B? Is this not the distance distribution between the green and red dots in Panel A? Is this not the bimodal distribution due to F656 flips in (red) and out (green) of the internal cavity? Maybe mark these peaks with something other than red and green dots or at least explain what the dots are doing in panel B.

Reviewer #2 (Remarks to the Author):

In their manuscript, Miranda et al. investigate the effect of ceramide lipids on voltage gating properties and K⁺ currents of the hERG1 potassium channel. The topic is very interesting, since the voltage gating in hERG, and especially its regulation by the lipid membrane is poorly understood; the recently established non-swapped topology of hERG makes the process even more interesting, suggesting that the regulation of this channel by lipids might be different than in non-swapped channels. Therefore, it is a well-timed study that eventually should be published in a well-renowned journal.

I will be not commenting much on the electrophysiological (experimental) parts of the paper, since it is not my expertise - I assume that these analyses were done correctly using state-of-the-art approaches. I will mostly focus on the computational aspects.

My main issue with the manuscript in the current form is the fact that, in my opinion, it lacks the connection between the computational part and the experimental one. The computational part revealed ceramide clustering around the voltage sensors and interactions with some residues based on contact analysis. However, the authors do not provide a mechanistic hypothesis how such a clustering would affect the motions of voltage sensors that eventually lead to the altered channel gating. Some residues interacting with ceramides in simulations affect voltage gating when mutated - this is very encouraging result, but again, the molecular mechanism of such effect would be highly desired. Therefore, in my view, the following points should be considered before the acceptance:

1. In Figure 5, a ceramide molecule is shown bound to the VSD, and the caption reads "Amphipathic ceramide molecules can flip-flop and bind to the VSD-PD interface of the channel (lipophilic route), enhancing both activation (opening) and deactivation (closure) of the channel." Do the authors have ideas how such a double function of ceramides might be achieved? Ideally one would like to see how 'jamming' of a ceramide molecule in between VSD and PD affects voltage gating ^[SEP].

2. Since the authors have the simulation systems for both open and closed conformational states, would it be possible to calculate whether binding of ceramides stabilize one of these states thermodynamically, via e.g. alchemical route? That could potentially greatly contribute to the penultimate paragraph in the Discussion ("...suggests that ceramide binding to the channel could be influenced by PD states.)

3. Is it possible that at least part of the ceramide effect on hERG gating is caused by the changes in the overall membrane properties induced by ceramides, instead of specific binding to the VSD? It

should be at least briefly mentioned in the discussion.

4. As mentioned, certain residues interacting with ceramides in MD turned out to modulate the effect of ceramides on voltage gating when mutated; did the authors check if these mutants show a normal voltage gating without ceramide application? In other words, is the effect of mutations limited to the interactions (or lack of thereof) with ceramides, or do they have other impact on voltage gating, not related to the ceramide application? As a negative control, it could be informative to mutate one of the residue from the VSD that do not interact with ceramides in MD, to check if it indeed does not have an effect on the ceramide impact on voltage gating.

5. In discussing the impact of ceramides on hERG, it might be helpful to compare the behavior of residues targeted by ceramides with their behavior in simulations done in membranes that do not contain ceramides (if I'm not mistaken, the authors performed such simulations for their previous work). For example, is the bimodal distribution of F656 sidechain specific to the systems containing ceramides? Similarly, the behavior of two major gating current contributors (K525 and R528) with and without ceramides can be of interest.

6. Since ceramides can apparently affect both activation and deactivation of hERG, and the authors found a number of residues that interact with ceramides only in the open state (page 4), wouldn't mutations of some of these residues affect only one of the two (i.e. activation or deactivation) potentially shedding more light on the molecular mechanism of ceramide action?

7. The authors suggest that ceramides target VSDs via flip-flopping in the membrane ("lipophilic route"). Is there any other evidence for this behavior than CG simulations? How quick would be this process (in reality) as compared to the hERG gating cycle? What is the distribution of ceramides in native membranes and is it asymmetric? If yes, what process / protein maintains such asymmetry? Would it compete with the flip-flop mechanism proposed here?

8. The flip-flop processes is inferred from the increased 'DXY' diffusion coefficient (although I find this symbol confusing) in Table 2 and an asymmetric density profile (SI Fig 2). It'd be good to actually show individual flip flop events and to estimate its rate directly from them. How does it compare with experimental estimates?

9. In the discussion / calculation of diffusion coefficients from MD simulations, did the authors consider corrections accounting for several effects, recently proposed by Gerhard Hummer and coworkers?

10. Even though individual simulations are rather long (10 us) the contact maps are not symmetric (Fig 3 C and D) even though the protein is; the author might want to discuss a potential lack of sampling and possible implications.

11. In the Introduction the authors write "Our experimental mutagenesis at the VSD-PD interface shows decreased *I_{hERG1}* block and reduced accumulation of ceramides in this region, respectively." I am not sure how the experimental mutagenesis show "reduced accumulation of ceramides in this region [VSD-PD]"? Can the authors explain?

Reviewer #3 (Remarks to the Author):

The manuscript presents results from a study aiming to shed light on molecular mechanisms responsible for the inhibitory effects of ceramides on hERG channels. The data include results from (1)-electrophysiological measurements on WT hERG1 and mutants expressed in mammalian HEK cells, and (2)-MD simulations of WT hERG1 with CG, AA, and combined CG-AA protocols in membranes containing a synthetic model of a ceramide lipid (named CER6) with C6 and C18 tails.

The results are used to relate observed interactions of CER6 at hERG residues, and the CER6 inhibition measured for hERG1 constructs mutated at those sites in order to infer on molecular mechanisms of hERG1 blockade by ceramides. Such inferences, based on the type of detailed functional and structural investigation utilized in this manuscript would be of great value to a variety of fields.

The manuscript is very well written and presents a level of detail that supports reproducibility and continuity of investigation. It is clear that much has been learned from the well thought-out simulations about sites and modes of molecular interaction with the CER6, and about the spectrum of inhibitory effects of CER6 in mutants.

Major concerns

1. The task of understanding functionally relevant interactions with the types of ceramides surrounding the channels under relevant physiological conditions is, of course, more complex than would appear from the manuscript. It is well known, for example, that the effects of ceramides on membrane protein function are strongly dependent on the ceramide type. A lengthening/shortening of the tail, or addition/removal of an unsaturation can change completely the nature or magnitude of the effect. It would seem essential for the manuscript to contain a documented discussion of this issue, and in particular of what types of ceramide are considered to be represented/not represented, by CER6. This would require at least an illustrative investigation of a ceramide identified to affect function using the same protocol, to establish the extent of similarity to CER6 data.

2. The mechanistic inferences discussed both in the Results and in the Discussion section are based on the identification of interactions sites and the behavior of the specific residues in the dynamics, and their consideration relative to results from inhibition in the corresponding mutants. But the manuscript contains no validation of the inference, e.g. by simulation of one or more mutant constructs. Absent this kind of validation, this type of prediction must assume that the mutation will simply eliminate (or enhance) the interaction without other effects. But Fig. 2A illustrates the diversity in the shapes of inhibition curves for the various mutants. Therefore, there is a need here for computational experiments examining effects of mutations that serve in the predictions, and especially for mutants F557L and F656C (See #3).

3. Related to point 2, above, there is a paucity of analysis and comment on the main experimental results in Fig. 2. The peculiar shapes of the inhibition curves for F557L and F656C is not even mentioned, nor is the fact that they do not achieve complete inhibition. In addition to providing a discussion of these points, it is necessary to include a quantitative evaluation of the validity of using IC50 values from such diverse curves (Fig. 2A) for the calculations described in the Legend of Fig. 2B, and perhaps reconsider the meaning/significance of the results shown in that panel.

4. The fact that the differences in curve shape and IC50 values are so large, when the mutation sites were identified in the manuscript just for being interaction sites with CER6, raises the mechanistic question of the mode in which CER6 (or the ceramides it represents), produces such diverse results. All the sites are predicted from the simulations to bind CER6, but the results show this to have dramatically different consequences. Therefore it becomes essential to discuss to what extent the manuscript contributes to the understanding of the “exact molecular mechanism of hERG1 blockade by ceramides” that was mentioned on p.1 (last line, left column).

RESPONSE TO REVIEWER COMMENTS AND SUMMARY OF REVISIONS

Reviewer #1 (Remarks to the Author):

Here the authors report insights into the molecular mechanism of ceramide inhibition of hERG channels. The authors find that ceramide shifts the activation curve to more hyperpolarized voltages, but decreases the current at depolarized voltages and decreases the open time. In simulations, they find that ceramide is concentrated at the interface between the voltage sensor domain and the pore domain and makes contact with residues at this interface. Mutations of some of these residues reduce the effect of ceramide, supporting a role for this interface for ceramide inhibition of hERG channels. I have only some comments about the presentation and details about the mechanism. The importance of hERG channels to human diseases makes this an important contribution to the field.

We thank the reviewer for the insightful comments and the enthusiasm about the work.

Pg. 4. The authors state that the mutations of T527 have no effect on ceramide-induced block, but what about effects on the ceramide-induced V shift? Or ceramide affinity? If you later want to claim that T527 is important for ceramide V shift, then you must show some data supporting that claim. Table 3 only shows $V_{0.5}$ due to mutation, not changes to ceramide-induced $V_{0.5}$ shifts.

Good point and we expanded our study to evaluate impact of the mutation in VSD domain to ceramide sensitivity. We assume that there is a typo in the reviewer's comment and the reviewer was referring to the T526X mutations (not T527). As suggested by the reviewer, we have performed new experiments to assess the effect of these mutations on channel's half-maximal activation voltage ($V_{1/2}^{act}$) before (control) and after application of [CER6] = 20 μ M (Supplementary Figure 2B) for various substitution in the position 526. The mutations significantly altering chemical nature of the side-chain (T526M and T526N) produce significant left-shift in the $V_{1/2}^{act}$ relative to WT and show differential response to the ceramide application. Specifically, ceramide application induced a significant right-shift of $V_{1/2}^{act}$ for hERG1A-T526M, but show no effects for hERG1A-T526N relative to control (Supplementary Figure 2B). The new experimental data highlight that the polarity and size of the sidechain at position 526 directly influences the effect of ceramides on $V_{1/2}^{act}$ in the channel. This agrees with our MD simulations, where the residue T526 was found to be specifically targeted by ceramides and not POPC lipids (Supplementary Figure 10).

Pg. 5. Not sure I understand the proposed mechanism? Why would mutations that have lower affinity for ceramide stabilize the open state (Tau increased) or vice versa? Pg. 5. And why would fast deactivating channels, such as hERG1B, have higher affinity for ceramide, if these contact residues are less exposed? Shouldn't it be the opposite? If a residue is less exposed then ceramide would have less opportunity to bind to this residue and the affinity should therefore be lower.

To further clarify the proposed mechanism of state-dependence in ceramide binding to hERG channel, we combined Free Energy Simulations (FES) performed on the hERG1 structures where the pore-domain in the open (Cryo-EM structure) and closed (EAG-based model) states with the electrophysiological evaluation of state-dependent activity and additional site-directed mutagenesis. To address Reviewer's concerns, we expanded our MD simulations to better map interaction/binding regions for ceramide. The aromatic residues present in the pore domain of hERG1 channel appears to modulate both the gating properties of the channel and the interactions of ceramides at the VSD-PD interface. We used contact-mapping to identify and then experimentally interrogate a number of single and double mutants for ceramide sensitivity. The new Fig. 3 provides direct evidence for pivotal role of the binding pocket identified in simulations for ceramide binding with double mutant (F656C/F557L), lacking ceramide sensitivity even at the highest ceramide concentration tested (500 μ M). Next, we performed FES calculations for ceramide bound to the pocket identified. The FEP results on the hERG1A-WT clearly show that ceramide has a higher affinity to the closed state of the pore-domain with binding free energy (ΔG) of -8.7 ± 1.6 kcal/mol vs -4.0 ± 1.1 for the open state. The new data on tonic blockade of hERG1A by ceramide application (new Fig. 3) are in agreement with the state-sensitive ceramide binding to the channel. See for example, new correlation analysis of $\Delta\tau^{deact}$ vs IC_{50}^{CER6} (summarized in new Fig. 3D). Therefore, we postulate that the mechanism emerging from the combined theory-experiment studies is best captured by the well-established concept of the conformational selection¹. We provide evidence that C₆ ceramide produces apparent tonic block of a putative closed or transitional state of the channel. Site-directed mutations that speed deactivation enhance the channels' likelihood of being in the closed state and thus available for block, thus producing block at lower

concentrations. Moreover, when deactivation is markedly slowed in selected mutations, the amount of open activity (residual current) even at the very highest concentrations testable is increased (see new Fig. 4 about the proposed mechanism). We are aware that the conceptual limitation to the conformational selection mechanism proposed in our study is related to the use of only two states in the theoretical designs e.g. open and closed PD with no account for a potential ceramide targeting to the transition states. In the framework of the conformational selection mechanism, the preferential stabilization of the closed-state of hERG1 PD by ceramides may reduce the probability of the open PD states (New Fig. 4). This would explain, at least in part, why hERG1A-variants (mutations introduced to the binding pocket at the VSD-PD interface) exhibit low sensitivity for ceramides (higher IC_{50}^{CER6}) and increased $\Delta\tau^{deact}$ (slower deactivation kinetics) compared to experiments performed for WT (Fig. 3D). The proposed mechanism would also explain the observed impact of ceramide application in the fast deactivating hERG1B and hERG1A-Y652A channels. The ceramide preferential binding to the closed pore-domain will further compliment rapid deactivation in these systems. Overall, we conclude that the ceramide-induced current inhibition observed in our electrophysiology experiments could result from decreased population of open-state conducting channels induced by ceramide addition (New Fig. 4).

Fig. 5. I don't get this model. B seems to violate micro-reversibility, but I think the two arrows mean different things? The left arrow probably means that the GV is shifted left, but the right arrow means that closing is faster. So the arrow have different meanings. A better model/picture is needed to better explain this in a self-consistent manner.

We thank the reviewer for this suggestion, it helps to clarify the message we try to deliver. Please see New Fig. 4 with explicit labelling and clear indication of the schematic nature for this figure + all arrows labeled as suggested.

Minor comments

Introduction. "results in activation of the channel at more depolarized potentials" should be at more hyperpolarized potentials.

Fixed.

Were the mutations introduced in Herg1A or Herg1B background? Also, please mention the difference in amino acid sequence or major structural difference between 1A and 1B.

Fixed. The revised text clearly states the major differences between isoforms in the section "Impact of mutations and isoform specificity on ceramide-induced effects" (left column, Pg. 4). The revised legend for New Fig. 3A indicates that the mutations were introduced in hERG1A background.

Please describe here what you mean by "contact duration" and "maximum occupancy" in simple words. It is not that difficult and it would make it a lot easier to read the rest of the text if this is explained here.

Thank you. The terminology has been clarified and the key concepts were explained in the section "Mapping ceramide contact regions for WT open- and closed- states" (right column, Pg. 3) e.g.

Contact duration: Sum of single-contact events.

Maximum occupancy: Longest single-contact event.

"Notably, T526 was specifically targeted by ceramides and not by POPC headgroups in CG-MD simulations due to the ability of ceramides to flip-flop, while R537 was accessible to both types of lipids (Supplementary Figure 8)." Only part of this is shown in Supplementary Fig 8. Fig 8 shows only that T526 is not contacted by POPC...

We now provide source data for new Fig. 2C (ceramide contacts in WT open-state) and new Supplementary Fig. 10 (POPC contacts). The analysis of lipid-specific occupancy clearly shows that residue T526 was involved in interactions with ceramides, but not with POPC lipids.

Do you mean "mutations of high occupancy residues"? Use this instead of "variants with substituted high occupancy residues".

Fixed.

Suppl. Fig 7. What does maximum single-contact event mean? You mean longest single-contact event? I would rename this for clarity.

The terminology “maximum single-contact event” was substituted by “Longest single-contact event” (see answer to question #3). See new Supplementary Figure 9.

Suppl Fig 7. "per ceramide molecule" could be misunderstood, which in my mind means an average (e.g. per capita income). I would call it "A histogram of the longest single-contact event for each ceramide molecules".

Great suggestion. Fixed. See new Supplementary Figure 9.

Suppl. Fig. 9A. Why is the amplitude 4? Due to 4 subunits? Or 4 ceramides interacting with one T526?

The red lines in the graph indicate the number of hydrogen bond pairs within 3.5 Å between the headgroup from a single ceramide molecule and the residue T526 from one of the VSDs. The upper limit for these interactions was found to be five and is indicated in the revised figure caption. See new Supplementary Figure 11.

Suppl Fig. 10. Why red and green dots in panel B? Is this not the distance distribution between the green and red dots in Panel A? Is this not the bimodal distribution due to F656 flips in (red) and out (green) of the internal cavity? Maybe mark these peaks with something other than red and green dots or at least explain what the dots are doing in panel B.

Thank you. We have made changes to this figure accordingly. See new Supplementary Figure 12.

Reviewer #2 (Remarks to the Author):

In their manuscript, Miranda et al. investigate the effect of ceramide lipids on voltage gating properties and K⁺ currents of the hERG1 potassium channel. The topic is very interesting, since the voltage gating in hERG, and especially its regulation by the lipid membrane is poorly understood; the recently established non-swapped topology of hERG makes the process even more interesting, suggesting that the regulation of this channel by lipids might be different than in non-swapped channels. Therefore, it is a well-timed study that eventually should be published in a well-renowned journal.

We thank the reviewer for the insightful comments and the positive evaluation of our work. We provide detailed summary of the revisions and point-by-point response below.

My main issue with the manuscript in the current form is the fact that, in my opinion, it lacks the connection between the computational part and the experimental one. The computational part revealed ceramide clustering around the voltage sensors and interactions with some residues based on contact analysis. However, the authors do not provide a mechanistic hypothesis how such a clustering would affect the motions of voltage sensors that eventually lead to the altered channel gating. Some residues interacting with ceramides in simulations affect voltage gating when mutated - this is very encouraging result, but again, the molecular mechanism of such effect would be highly desired. Therefore, in my view, the following points should be considered before the acceptance:

Thank you and we have attempted to improve on the cohesion between experimental and theoretical designs of the study. Please also see our response to the Reviewer 1. We have made sure that the new Figs 2, 3 and 4 provide a clear message on the intimate coupling between simulation and experimental data. Specifically, through the generation of testable hypothesis from mapping out key elements of ceramide binding to hERG1 in different states and their experimental validation. Taken together, data from modelling and experiment led us to the proposed mechanism of conformational selection, explaining intricacies of the hERG1 blockade by ceramides. We also attempted to outline limitations of the approach in the revised manuscript to stimulate further explorations by the field.

In Figure 5, a ceramide molecule is shown bound to the VSD, and the caption reads “Amphipathic ceramide molecules can flip-flop and bind to the VSD-PD interface of the channel (lipophilic route), enhancing both activation (opening) and deactivation (closure) of the channel.” Do the authors have ideas how such a double function of ceramides might be achieved? Ideally one would like to see how ‘jamming’ of a ceramide molecule in between VSD and PD affects voltage gating.

With the new experimental data on T526X mutants we have sufficient evidence to argue that the specific targeting of T526 by ceramides (New Supplementary Figure 2B) could drive, at least in part, the activation of hERG1-variants at more hyperpolarized membrane potentials. This residue is in close proximity of key gating charges in S4 K525 and R528 (Fig. 1D) of hERG1². The T526S mutant shows no ceramide sensitivity in the $V_{1/2}^{act}$ (New Supplementary Figure 2B), highlighting the critical importance of the hydroxyl group at this position for ceramide (polar) interactions at the VSD. This is in excellent agreement with MD simulations results (Supplementary Figure 10). Overall, new experiments show that the polarity and size of the sidechain at position 526 directly influences the effect of ceramides on $V_{1/2}^{act}$ in the channel (see reply to Q1 for Reviewer #1). Unfortunately, the structures of the VSD captured in Cryo-EM for hERG1 and hEAG1 channels are almost identical^{3,4}, preventing direct modelling of the S4 movement to the hyper-polarized (closed) states. The structural mechanism connecting ceramide binding to the pocket formed by VSD and PD and the sensor movement will remain a next frontier to us and the field.

As suggested by the Reviewer, we have expanded analysis of the flip-flop events involving ceramides. The analysis of the flip-flop events in microseconds-long CG-MD simulations of POPC:CER6 membrane patch is provided in the updated Table 2 and new Supplementary Figure 3. We would like to note however, that the lack of VSDs in the hyperpolarized state in our closed-state model (discussed above) hampers a thorough analysis on the role of ceramide flip-flop during channel gating transitions. We apologize for the promoting a potentially misleading interpretations of the role played by the flip-flop events, and revised caption of new Fig. 4.

Since the authors have the simulation systems for both open and closed conformational states, would it be possible to calculate whether binding of ceramides stabilize one of these states thermodynamically, via e.g. alchemical route? That

could potentially greatly contribute to the penultimate paragraph in the Discussion (“...suggests that ceramide binding to the channel could be influenced by PD states.”)

Excellent suggestion! Following, the reviewer advise we have now added an exhaustive mapping of the ceramide binding site constituents for ceramide and identified a double mutant (F656/F557L) that renders channel to be virtually ceramide-insensitive (new Fig. 3). Once the identity of the binding pocket residues in PD has been confirmed, we performed FES simulations of state-dependent binding of ceramide to open- and closed-states of PD. The additional FES data clearly indicate the preferential binding to the closed state and hence hints the conformational selection mechanism. To test this exciting development, we have performed tonic block measurement and confirmed preferential binding to the closed state of the pore domain. The combined evidence from electrophysiology experiments and FEP calculations suggests that the closed-state PD is stabilized by ceramide binding to the VSD-PD interface, close to the location of aromatic residues F557L and F656. We propose that the stabilization of the closed-state PD by ceramides leads to the observed current inhibition through a conformational selection mechanism via a lipophilic route (new Fig. 4).

Is it possible that at least part of the ceramide effect on hERG gating is caused by the changes in the overall membrane properties induced by ceramides, instead of specific binding to the VSD? It should be at least briefly mentioned in the discussion.

We agree with the reviewer that changes in the overall membrane properties may, at least partially, contribute to the observed effect. The discussion of additional components of the ceramide-induced blockade is added to feature previous studies putting forward a mechanism involving lipid-raft formation⁵. The authors⁵ used electrophysiology to propose a lipid-raft hypothesis for hERG1 modulation by ceramides. The discussion in Ref⁵ indicates that the lipid-raft hypothesis is insufficient to explain many of the voltage-dependent features present in the ceramide-induced I_{hERG1} inhibition. Furthermore, CER6 has shown to disrupt the formation of lipid rafts in supported lipid bilayers composed of DOPC, sphingomyelin and cholesterol⁶ and hence, role of the lipid rafts in ceramide-induced inhibition of hERG1 remains debatable. Other mechanisms rely on the impact of the reactive oxygen species (ROS) on the channel function inhibition and altered gating kinetics in hERG1^{7,8}. Both mechanisms, however, were found to be highly dependent on the experimental conditions and protocols, although they should not be completely ruled out^{5,7,8}. Our new data clearly shows a state-dependence of hERG1 blockade both in FES and tonic blockade measurements, which combined with the lack of CER6 inhibition of F656C/F557L mutant puts forward mechanism of direct and specific channel targeting by ceramides.

As mentioned, certain residues interacting with ceramides in MD turned out to modulate the effect of ceramides on voltage gating when mutated; did the authors check if these mutants show a normal voltage gating without ceramide application? In other words, is the effect of mutations limited to the interactions (or lack of thereof) with ceramides, or do they have other impact on voltage gating, not related to the ceramide application?

Please see our response to the Reviewer 1. The revised text, figures and tables provide comparison of the mutation-induced changes in channel kinetics with and without ceramides.

As a negative control, it could be informative to mutate one of the residue from the VSD that do not interact with ceramides in MD, to check if it indeed does not have an effect on the ceramide impact on voltage gating.

Thank you – very interesting idea. To follow on the reviewer’s suggestion, we experimentally tested the L529A mutation in the S4 helix. MD simulations showed that L529 is not participating in interactions with ceramides. The electrophysiology of hERG1-L529A demonstrate lack of ceramide-induced impacts on $V_{1/2}^{act}$ and τ^{deact} relative to controls (Supplementary Figure 2B and C). Furthermore, WT and L529A systems exhibited statistically similar blockade (%) of hERG1 current (Supplementary Figure 2A).

In discussing the impact of ceramides on hERG1, it might be helpful to compare the behavior of residues targeted by ceramides with their behavior in simulations done in membranes that do not contain ceramides (if I’m not mistaken, the authors performed such simulations for their previous work). For example, is the bimodal distribution of F656 sidechain specific to the systems containing ceramides? Similarly, the behavior of two major gating current contributors (K525 and R528) with and without ceramides can be of interest.

To better assess the influence of ceramides (or lack of thereof) on the dynamics of the specific residues mentioned by the reviewer, we performed additional analysis using a previously run two microseconds-long unbiased AA-MD simulation of hERG1 inserted in pure POPC membrane patch as control⁹. The comparison of ceramide-free and ceramide-present all-atom simulation indicate a subtle impact of the CER6 on the distribution of the conformational states for F656 sidechain (New Supplementary Figure 12B). The presence of ceramides in the membrane promotes the F656 orientation towards the lipid-facing interface, whereas the ceramide-free simulations indicate a higher probability of pore-facing orientation of F656. Overall, these results suggest that F656, besides modulating channel deactivation kinetics as observed in our mutagenesis experiments, could potentially be involved in direct interactions with ceramides. Although we did not observe direct interactions between F656 and ceramides in our CG-MD simulations (see Source Data for Fig. 2 C,D), the double mutation F656C/F557L markedly reduced ceramide binding into the VSD-PD interface of the channel. This highlights the pivotal role of F656 in mediating hERG1A-ceramide interactions (either directly or indirectly). As suggested by the reviewer, we analyzed the dynamics of K525 and R528 located in the S4-helix of the channel (Fig. 1D). AA-MD simulations performed in pure POPC or POPC:CER6 showed almost identical distribution of states for these residues.

Since ceramides can apparently affect both activation and deactivation of hERG, and the authors found a number of residues that interact with ceramides only in the open state (page 4), wouldn't mutations of some of these residues affect only one of the two (i.e. activation or deactivation) potentially shedding more light on the molecular mechanism of ceramide action?

Please see our response to the Reviewer 1. The revised text, figures and tables provide comparison of the mutation-induced changes in channel kinetics and current inhibition, with and without ceramides.

The authors suggest that ceramides target VSDs via flip-flopping in the membrane ("lipophilic route"). Is there any other evidence for this behavior than CG simulations? How quick would be this process (in reality) as compared to the hERG gating cycle? What is the distribution of ceramides in native membranes and is it asymmetric? If yes, what process / protein maintains such asymmetry? Would it compete with the flip-flop mechanism proposed here?

We reassessed the role of ceramide flip-flop and revised the language used to describe the mechanism (see discussion in the last paragraph of our reply to Q1). Experimentally, ceramides were applied from the extracellular solution and hence, an asymmetric distribution as pointed out by the Reviewer is entirely feasible. We toned this discussion down, accordingly.

The flip-flop processes is inferred from the increased 'DXY' diffusion coefficient (although I find this symbol confusing) in Table 2 and an asymmetric density profile (SI Fig 2). It'd be good to actually show individual flip flop events and to estimate its rate directly from them. How does it compare with experimental estimates?

The reviewer is correct that the paper has been focusing on the lateral ceramide diffusion. We have clarified definition of D_{x-y}^{lipid} in the revised manuscript. As suggested, we provide now analysis of the flip-flop events from the coarse-grained POPC:CER6 bicomponent membrane. A total of 18 flip-flop events for 14 (out of 100) CER6 molecules were reported (revised Table 2) corresponding to rate of $3.6 \pm 1.4 \times 10^6 \text{ s}^{-1}$ (Supplementary Figure 2E). This rate is two orders of magnitude faster than previously reported for longer chain C₁₈- and C₂₄- ceramides ($2.0 \pm 0.4 \times 10^4 \text{ s}^{-1}$) in multi-microseconds long CG-MD simulation of a realistic plasma membrane¹⁰. A faster rate for the short chain C₆-ceramide is in agreement with experimental data¹¹. The experimental flip-flop rates for short chain ceramides (C₅, C₆) have been reported^{11,12} in the range of $\sim 10^4 \text{ s}^{-1}$ to 10 s^{-1} with high dependence on the methodologies, membrane models and conditions. The increased flip-flop rates in CG-MD vs AA-MD (not observed) and experiments are a consequence of the smoother energy surface in CG systems, which enhances the sampling compared to AA-MD simulations^{10,13,14}.

In the discussion / calculation of diffusion coefficients from MD simulations, did the authors consider corrections accounting for several effects, recently proposed by Gerhard Hummer and coworkers?

We outlined the methodology used for the calculation of diffusion coefficients in the Methods section. This methodology is the same used by Ingolfsson *et al.*¹⁰ to study the lateral diffusion of lipids in CG-MD simulations of a realistic membrane. Briefly, all calculations relied on the linear fit of the MSD vs. time dependencies. We have not used auto-correlation function analysis nor we adopted development in the diffusion coefficient estimators reported by Gerhard Hummer and colleagues¹⁵. Our main goal was to qualitatively assess ceramide dynamics in a model membrane. We

found that the MSD-time dependence from CG simulations is rigorously described with a linear fit of MSD over the time-scale studied. The recent (2020) GLS estimator of Hummer and colleagues showed an improved performance for the non-linear regions present in the MSD vs. time function (anomalous diffusion regions), which were not present in our case.

Even though individual simulations are rather long (10 us) the contact maps are not symmetric (Fig 3 C and D) even though the protein is; the author might want to discuss a potential lack of sampling and possible implications.

We agree with the reviewer and now explicitly discuss sampling limitations (first paragraph, right column, page 5). Despite this challenge common to all MD simulations, we still conclude that MD and the added FEP simulations provided testable hypothesis for the experiment.

*11. In the Introduction the authors write “Our experimental mutagenesis at the VSD-PD interface shows decreased *h*ERG1 block and reduced accumulation of ceramides in this region, respectively.” I am not sure how the experimental mutagenesis show “reduced accumulation of ceramides in this region [VSD-PD]”? Can the authors explain?*

We apologize for the confusion. The text has been revised accordingly to indicate that we have used analysis of the CG-MD simulations to design mutants with the reduced ceramide occupancies (F656/F557L). This double mutant experimentally shows very little sensitivity to application of ceramide. We changed this statement to: “*In-silico* mutagenesis at the VSD-PD interface shows reduced accumulation of ceramides in the region.”

Reviewer #3 (Remarks to the Author):

The manuscript presents results from a study aiming to shed light on molecular mechanisms responsible for the inhibitory effects of ceramides on hERG channels. The data include results from (1)-electrophysiological measurements on WT hERG1 and mutants expressed in mammalian HEK cells, and (2)-MD simulations of WT hERG1 with CG, AA, and combined CG-AA protocols in membranes containing a synthetic model of a ceramide lipid (named CER6) with C6 and C18 tails. The results are used to relate observed interactions of CER6 at hERG residues, and the CER6 inhibition measured for hERG1 constructs mutated at those sites in order to infer on molecular mechanisms of hERG1 blockade by ceramides. Such inferences, based on the type of detailed functional and structural investigation utilized in this manuscript would be of great value to a variety of fields. The manuscript is very well written and presents a level of detail that supports reproducibility and continuity of investigation. It is clear that much has been learned from the well thought-out simulations about sites and modes of molecular interaction with the CER6, and about the spectrum of inhibitory effects of CER6 in mutants.

We thank the reviewer for the insightful comments and the positive evaluation of our work. We provide detailed summary of the revisions and point-by-point response below.

Major concerns

The task of understanding functionally relevant interactions with the types of ceramides surrounding the channels under relevant physiological conditions is, of course, more complex than would appear from the manuscript. It is well known, for example, that the effects of ceramides on membrane protein function are strongly dependent on the ceramide type. A lengthening/shortening of the tail, or addition/removal of an unsaturation can change completely the nature or magnitude of the effect. It would seem essential for the manuscript to contain a documented discussion of this issue, and in particular of what types of ceramide are considered to be represented/not represented, by CER6. This would require at least an illustrative investigation of a ceramide identified to affect function using the same protocol, to establish the extent of similarity to CER6 data.

Thank you for excellent suggestion. We have experimentally investigated inhibitory effects of various ceramides on hERG1 current. We considered now C₂, C₆ and C₈ to evaluate impact of the tail length and provide this new data in New Supplementary Fig. 1A. The choice of ceramides was dictated by cell-membrane permeability⁶. Application of C₈ resulted in the IC₅₀^{CER8} and Hill coefficient of ~98 uM and ~1, respectively, indicating a five-fold decrease of inhibitory potency in respect to C₆. To the best of our knowledge, this is the first time that C₈ ceramide is tested for inhibitory

activity against hERG1. For C₂, we were unable to reach concentrations required for accurate IC_{50}^{CER2} estimates even at 500 μ M (New Supplementary Fig. 1A). This is in agreement with the work of Bai *et al.*⁷. Therefore, C₆ ceramide is the strongest channel blocker among all of the tested ceramides in this work. These results indicate that the shortening/lengthening of ceramide hydrophobic tail is one of the primary factors in control of the inhibitory effects on hERG1. This finding provides further evidence for the importance of hydrophobic interactions between ceramide tail and the residues at VSD-facing surface of the PD in hERG1.

The mechanistic inferences discussed both in the Results and in the Discussion section are based on the identification of interactions sites and the behavior of the specific residues in the dynamics, and their consideration relative to results from inhibition in the corresponding mutants. But the manuscript contains no validation of the inference, e.g. by simulation of one or more mutant constructs. Absent this kind of validation, this type of prediction must assume that the mutation will simply eliminate (or enhance) the interaction without other effects. But Fig. 2A illustrates the diversity in the shapes of inhibition curves for the various mutants. Therefore, there is a need here for computational experiments examining effects of mutations that serve in the predictions, and especially for mutants F557L and F656C (See #3).

Thank you for pointing it out. For this resubmission, we performed additional FES calculations, identified additional spots for mutagenesis and provided proof-of-principle modelling and experimental data to the proposed mechanisms. Specifically, our new data obtained from the analysis of CG MD simulations helped to identify key binding pocket and then test role of specific side-chain chemistry in ceramide binding. We show in the revised submission that a double mutations F656C/F557L virtually obliterates channel sensitivity to ceramides. Please also see our responses to the questions raised by the Reviewers 1 and 2. FES data clearly indicates state-dependence of ceramide binding to open- and closed- conformations of PD. The analysis of tonic block confirmed state-dependence of the block. Figure 2 and Figure 3 have been completely revised to better present the mechanism and provide additional experimental evidence. We performed various control wet-lab and computational experiments in support of the mechanism present in the revised Fig. 4.

Related to point 2, above, there is a paucity of analysis and comment on the main experimental results in Fig. 2. The peculiar shapes of the inhibition curves for F557L and F656C is not even mentioned, nor is the fact that they do not achieve complete inhibition. In addition to providing a discussion of these points, it is necessary to include a quantitative evaluation of the validity of using IC50 values from such diverse curves (Fig. 2A) for the calculations described in the Legend of Fig. 2B, and perhaps reconsider the meaning/significance of the results shown in that panel. The fact that the differences in curve shape and IC50 values are so large, when the mutation sites were identified in the manuscript just for being interaction sites with CER6, raises the mechanistic question of the mode in which CER6 (or the ceramides it represents), produces such diverse results. All the sites are predicted from the simulations to bind CER6, but the results show this to have dramatically different consequences. Therefore it becomes essential to discuss to what extent the manuscript contributes to the understanding of the “exact molecular mechanism of hERG1 blockade by ceramides” that was mentioned on p.1 (last line, left column).

Thank you. The Fig. 2 the reviewer is referring to, has been revised completely (see new Fig 3). We also provide now a correlation analysis based on the raw data from the electrophysiology (new Fig. 3) in support of the mechanisms proposed in new Fig. 4. In general, we found a strong correlation between $\Delta\tau^{deact}$ and IC_{50}^{CER6} (new Fig. 3D), indicating that the inhibitory potency of ceramides will depend on the changes in channel's transition rate towards the closed state induced by ceramides vs control. Our conformational selection mechanism (new Fig. 4) could help explain the presence of residual currents for hERG1A- F557L, F656C and F56C/F557L, which display slower deactivation rates. Since ceramides does not manifest an apparent open-state block, probabilistically some of the channels remain unblocked and conducting potassium currents. Our new data also indicates that ceramide produces tonic block, suggesting a high probability of closed state block without evidence for use-dependent block. This agrees with the near-absence of residual currents for fast deactivating channels hERG1B and hERG1A-Y652A with respect to the other hERG1A-variants. In this resubmission, we avoided the use of the phrase “exact mechanism”. The correlation found between $\Delta\tau^{deact}$ and IC_{50}^{CER6} , and the effects of mutations at the VSD-PD interface on channel inhibition and gating kinetics indicates a general trend.

References

- 1 Hammes, G. G., Chang, Y. C. & Oas, T. G. Conformational selection or induced fit: a flux description of reaction mechanism. *Proc Natl Acad Sci U S A* **106**, 13737-13741, doi:10.1073/pnas.0907195106 (2009).
- 2 Zhang, M., Liu, J. & Tseng, G. N. Gating charges in the activation and inactivation processes of the HERG channel. *J Gen Physiol* **124**, 703-718, doi:10.1085/jgp.200409119 (2004).
- 3 Wang, W. & MacKinnon, R. Cryo-EM structure of the open human ether-a-go-go-related K⁺ channel HERG. *Cell* **169**, 422-430, doi:10.1016/j.cell.2017.03.048 (2017).
- 4 Whicher, J. R. & MacKinnon, R. Structure of the voltage-gated K(+) channel Eag1 reveals an alternative voltage sensing mechanism. *Science* **353**, 664-669, doi:10.1126/science.aaf8070 (2016).
- 5 Ganapathi, S. B., Fox, T. E., Kester, M. & Elmslie, K. S. Ceramide modulates HERG potassium channel gating by translocation into lipid rafts. *Am J Physiol Cell Physiol* **299**, C74-86, doi:10.1152/ajpcell.00462.2009 (2010).
- 6 Chiantia, S., Kahya, N. & Schwille, P. Raft domain reorganization driven by short- and long-chain ceramide: a combined AFM and FCS study. *Langmuir* **23**, 7659-7665, doi:10.1021/la7010919 (2007).
- 7 Bai, Y. *et al.* Sphingolipid metabolite ceramide causes metabolic perturbation contributing to HERG K⁺ channel dysfunction. *Cell Physiol Biochem* **20**, 429-440, doi:10.1159/000107527 (2007).
- 8 Chapman, H. *et al.* Downregulation of the HERG (KCNH2) K(+) channel by ceramide: evidence for ubiquitin-mediated lysosomal degradation. *J Cell Sci* **118**, 5325-5334, doi:10.1242/jcs.02635 (2005).
- 9 Miranda, W. E. *et al.* Selectivity filter modalities and rapid inactivation of the hERG1 channel. *Proc Natl Acad Sci U S A* **117**, 2795-2804, doi:10.1073/pnas.1909196117 (2020).
- 10 Ingolfsson, H. I. *et al.* Lipid organization of the plasma membrane. *J. Am. Chem. Soc.* **136**, 14554-14559, doi:10.1021/ja507832e (2014).
- 11 Lopez-Montero, I. *et al.* Rapid transbilayer movement of ceramides in phospholipid vesicles and in human erythrocytes. *J Biol Chem* **280**, 25811-25819, doi:10.1074/jbc.M412052200 (2005).
- 12 Bai, J. & Pagano, R. E. Measurement of spontaneous transfer and transbilayer movement of BODIPY-labeled lipids in lipid vesicles. *Biochemistry* **36**, 8840-8848, doi:10.1021/bi970145r (1997).
- 13 Marrink, S. J., de Vries, A. H. & Mark, A. E. Coarse grained model for semiquantitative lipid simulations. *J Phys Chem B* **108**, 750-760, doi:10.1021/jp036508g (2004).
- 14 Marrink, S. J., Risselada, H. J., Yefimov, S., Tieleman, D. P. & de Vries, A. H. The MARTINI force field: coarse grained model for biomolecular simulations. *J Phys Chem B* **111**, 7812-7824, doi:10.1021/jp071097f (2007).
- 15 Bullerjahn, J. T., von Bulow, S. & Hummer, G. Optimal estimates of self-diffusion coefficients from molecular dynamics simulations. *J Chem Phys* **153**, 024116, doi:10.1063/5.0008312 (2020).

Reviewer #1 (Remarks to the Author):

The authors have responded well to my comments. However, there are some things that need to be fixed.

1. Fig 2A and Discussion around this. To me this is not tonic block. In Fig 2B it is clearly shown that the current in response to the first voltage pulse (not tail voltage pulse) is not reduced, and it is actually increased (maybe not significant?) for the first pulse. And for every pulse afterwards, the current initially is the same as in control. Instead, the block occurs during the pulse, kind of like the kinetics of a slow open-state blocker that falls off during hyperpolarizations. So there is no evidence of tonic block, which should reduce the initial currents of the first voltage pulse. To me, this suggests that the block/inhibition is not in the closed state at the resting membrane potential, but something that occurs closer to the open state. Maybe in a preactivated closed state with S4 up and the gate closed. This is actually the model you have for the closed state. So why not go with that one? To me, the data suggest that ceramide binds to the open state and forces the channel closed when S4 is still up. This would then explain why channels with less stable open state binds ceramide better...

2. Your FEB calculation suggests that ceramide binds better to this preactivated closed state than the open state. Why not show or mention the residues involved in the binding of ceramide in the closed state, instead of listing all residues that bind in the open state but not in the closed state ("We found that residues T526, V535, R537, Y542, and Y545 (S4 helix); F551, I560, and I567 (S5 helix), and N658 (S6 helix) are involved in the formation of maximum-occupancy binding pockets in the open-state of hERG1-WT but not for the closed-state"). This seems more important to know the closed state residues than the exclusive open state residues.

3. Fig 3D&E. These correlations do not make sense to me. It seems to me that you want to correlate the ceramide affinity with the tau and $V_{0.5}$ of the channel mutants, not with the changes induced by ceramide, if you want to support your model of less strong binding to the open state.

4. Discussion. "The lack of effect on $V_{0.5}$ after ceramide application for T526S mutant indicates that the hydroxyl group at this position plays a major role for ceramide (polar) interactions ". This, to me, is not clearly written...it is the lack of ceramide effect on other T526 mutants (but still ceramide effect on T526S) that highlights the role of the OH group...

Reviewer #2 (Remarks to the Author):

The authors have sufficiently address all my comments, added new experiments and analyses and markedly improved the manuscript. I have only two minor comments before I recommend the manuscript for acceptance:

1. I am not sure if the red cross over the ceramide molecule in Fig 4 is justified. Free energy calculations show that ceramide binds to the open state with dG of ~ -4.0 kcal/mol, so I suppose the molecule is still expected to be there?

2. Through free energy calculations, the authors have now a free energy value by which C6 stabilizes the closed state with respect to the open state (using a thermodynamic cycle that would be 4.7

kcal/mol. Why not report it like this?

Reviewer #3 (Remarks to the Author):

The detailed responses and the corresponding additions and revisions in the manuscript provide very reasonable support and argumentation for the major findings reported in the revised manuscript. The connection and the interplay between the experiments and findings from electrophysiology and MD simulations approaches is clearly demonstrated and proven by the arguments and the manuscript to be necessary for the interpretation results from either one. This has been clarified further by the changes and additions in the revision.

The manuscript was enriched with relevant new data and support for assertions with appropriate modifications, as well as an improved discussion. Thus, the added mutation data and the corresponding FEP calculations have answered many of the questions raised by my own evaluation and those of other reviewers. The description in the new Fig. 4 does clarify the mechanistic conclusions put forth by the authors, even if their classification of the mechanism in the “selection vs induced fit” frame is not as complete, nor as definitive as assumed. It is nevertheless clear and hence appropriate for further investigation and refinement.

In my view the work presented in the revised manuscript constitutes a significant advance in this field, the voluminous data are valid as obtained and valuable as a base of information, and the conclusions are presented with sufficient clarity to be intriguing and inviting of further investigative steps.

RESPONSE TO REVIEWER COMMENTS

We would like to thank the Reviewers for very constructive suggestions. They really helped to improve the manuscript.

Reviewer #1 (Remarks to the Author):

The authors have responded well to my comments. However, there are some things that need to be fixed.

1. Fig 2A and Discussion around this. To me this is not tonic block. In Fig 2B it is clearly shown that the current in response to the first voltage pulse (not tail voltage pulse) is not reduced, and it is actually increased (maybe not significant?) for the first pulse. And for every pulse afterwards, the current initially is the same as in control. Instead, the block occurs during the pulse, kind of like the kinetics of a slow open-state blocker that falls off during hyperpolarizations. So, there is no evidence of tonic block, which should reduce the initial currents of the first voltage pulse. To me, this suggests that the block/inhibition is not in the closed state at the resting membrane potential, but something that occurs closer to the open state. Maybe in a preactivated closed state with S4 up and the gate closed. This is actually the model you have for the closed state. So why not go with that one? To me, the data suggest that ceramide binds to the open state and forces the channel closed when S4 is still up. This would then explain why channels with less stable open state binds ceramide better...

We appreciate the Reviewer comment and agree that the data shown in the initial submission was insufficient to clearly distinguish use-dependent and tonic block of hERG1 by ceramide. One confounding factor is the use of HEK cells in the initial experimental design. While this cell line allows for the efficient transfection with various hERG1 variants, it also shows significant endogenous current. To attain “cleaner”, low-noise recordings, we performed new experiments to evaluate tonic vs use-dependent block (see new Fig. 2a, b) with stable-expressing hERG1-WT cells derived from human cardiomyocytes¹. This cell-line has the advantage of displaying very little endogenous currents (e.g. near complete lack of an inactivating I_{to}). The new Figs. 2a/b clearly shows a marked reduction of the currents elicited by the pulse P₁, with no incremental block between P₂₋₁₅. In addition, the time-dependent currents show no use-dependent features and this is paralleled by the tail currents. The tonic-block is also in line with our free-energy calculations that show more favourable binding for the closed state pore structure. These data taken in concert, support the notion of apparent ceramide-induced tonic block. The interpretation of this apparent tonic block suggests that ceramide targets a closed state or pre-open (preactivated) state of the channel. However, although we observe no evidence for the use-dependent block, we cannot rule out very rapid time-dependent block as pointed by the Reviewer, that is beyond our ability to register on the instrument resolution used in this study. The revised text on the page 5 introduces a possibility of the mechanism suggested by the Reviewer.

In summary, we cannot completely discard the possibility of ceramide binding to transition or pre-activated states of the channel, as pointed by the reviewer. However, our tonic block experiments cannot distinguish between the “true” closed state with both PD and VSD in the resting (closed) conformations and presumably pre-activated state captured in the Cryo-EM structure from EAG channel² used to model hERG1 closed-state PD. It can be concluded from our experiments and simulations, however, that ceramide does exhibit preferential binding to the PD in its closed state. Thus, our new data corroborate the initially proposed mechanism, but we revised the text to outline potential limitations, alternative explanations and clearly linked it to the states captured in the Cryo-EM structures.

2. Your FEB calculation suggests that ceramide binds better to this preactivated closed state than the open state. Why not show or mention the residues involved in the binding of ceramide in the closed state, instead of listing all residues that bind in the open state but not in the closed state (“We found that residues T526, V535, R537, Y542, and Y545 (S4 helix); F551, I560, and I567 (S5 helix), and N658 (S6 helix) are involved in the formation of maximum-occupancy binding pockets in the open-state of hERG1-WT but not for the closed-state”). This seems more important to know the closed state residues than the exclusive open state residues.

Thank you – excellent suggestion. In this submission, we list all of the interacting partners as suggested. See the following sentence in the revised version: “We found apolar and polar residues at the VSD-PD interface involved in the formation of maximum-occupancy binding pockets in the closed-state of hERG1-WT (see Source Data – Figure 2c, d), specifically L532, V533, L539, E544 (S4 helix); A561, H562, W563 (S5 helix); Y611, V612, I639, L666, Y667 (S6 helix). This is in line with the amphipathic nature of ceramide molecules.”

3. Fig 3D&E. These correlations do not make sense to me. It seems to me that you want to correlate the ceramide affinity with the tau and $V_{0.5}$ of the channel mutants, not with the changes induced by ceramide, if you want to support your model of less strong binding to the open state.

We agree with the reviewer that the correlation was present in Fig. 3d,e may lead to a confusion and they do not contribute to an understanding of cause-effect relationships. Indeed, we are not expecting linear relationships between pIC50 and the above-mentioned gating parameters due to complexities inherent to the proposed allosteric mechanism. Therefore, we decided to remove any correlation metrics/or line of the best fit in Fig. 3d,e in the revised submission (revised Figure 3).

We prefer to show data reflecting the $\Delta\tau_{deact}$ and $\Delta V_{0.5}^{act}$ after ceramide application relative to the controls ($\Delta = ctrl - ceramide$) to qualitatively assess the variation of these kinetic parameters for a series of experiments. However, the reviewer is correct and presenting actual values after ceramide application may be useful for the readers. In the revised version, we include the plots for pIC50 vs $\tau_{deact/cer}$ and pIC50 vs $V_{0.5/cer}^{act}$ after ceramide application as part of the supplementary information (Supplementary Fig. 13a, b). Overall, the graphs in Fig. 3d,e are intended to show the trends among ceramide affinity (pIC50) and the changes in gating parameters after ceramide application, and not to justify our proposed conformational selection mechanism where ceramides bind preferentially to the closed state. Rather, the preferential binding of ceramides to the closed-state is justified by our new tonic block electrophysiology experiments (see answer to question 1) and also supported by our free-energy calculations.

In retrospect, the most important criticism of the correlation analyses is that both properties, speeding of deactivation and shifts in activation $V_{0.5}$ are occurring simultaneously. Therefore, the cause-effect relationships cannot be clearly evaluated for each property alone. Accordingly, we believe that the best solution might be to simply remove this correlation analyses from the manuscript, but retain discussion of the qualitative and apparent trends. Even so, we wish to present the data as requested by the reviewer.

We also present this data in the main text and in the Supplementary Information to the community of scholars.

4. Discussion. “The lack of effect on $V_{0.5}$ after ceramide application for T526S mutant indicates that the hydroxyl group at this position plays a major role for ceramide (polar) interactions “. This, to me, is not clearly written...it is the lack of ceramide effect on other T526 mutants (but still ceramide effect on T526S) that highlights the role of the OH group...

We agree with the reviewer and re-phrased the sentence as follows: “Only the S526 substitution (among T526X mutants) shows ceramide-induced left-shifting of $V_{1/2}^{act}$ statistically similar to WT. This indicates that the hydroxyl group at this position plays an important role in ceramide (polar) interactions at the VSD as suggested by our MD simulations.”

Reviewer #2 (Remarks to the Author):

The authors have sufficiently addressed all my comments, added new experiments and analyses and markedly improved the manuscript. I have only two minor comments before I recommend the manuscript for acceptance:

1. I am not sure if the red cross over the ceramide molecule in Fig 4 is justified. Free energy calculations show that ceramide binds to the open state with dG of ~ -4.0 kcal/mol, so I suppose the molecule is still expected to be there?

We agree with the Reviewer and revised Figure 4 accordingly. The cross has been removed and the states are labeled now as “Lower-affinity” and “Higher-affinity” to better reflect experimental and computational results of the study.

2. Through free energy calculations, the authors have now a free energy value by which C6 stabilizes the closed state with respect to the open state (using a thermodynamic cycle that would be 4.7 kcal/mol). Why not report it like this?

Excellent suggestion. We added the following sentence in the last paragraph of the section **Mapping ceramide contact regions for WT open- and closed- PD states** (Pg 4, left column): “This indicates that WT-ceramide complex is around 4.7 kcal/mol more stable for the closed- vs open- state PD”.

Reviewer #3 (Remarks to the Author):

In my view the work presented in the revised manuscript constitutes a significant advance in this field, the voluminous data are valid as obtained and valuable as a base of information, and the conclusions are presented with sufficient clarity to be intriguing and inviting of further investigative steps.

Thank you so much and we really appreciate time and effort of the Reviewer.

References

- 1 Bhat, R., Houghton, M. Recombinant cardiomyocytes and cardiomyocyte cell lines expressing herg. USA patent (2020).
- 2 Whicher, J. R. & MacKinnon, R. Structure of the voltage-gated K(+) channel Eag1 reveals an alternative voltage sensing mechanism. *Science* **353**, 664-669 (2016).

Reviewer #1 (Remarks to the Author):

The authors have responded well to my comments